# EverybodyDance: Bipartite Graph–Based Identity Correspondence for Multi-Character Animation

**Haotian Ling**[1,2*]   **Zequn Chen**[2†]   **Qiuying Chen**[3]

**Donglin Di**[2]   **Yongjia Ma**[2]   **Hao Li**[2]   **Chen Wei**[2]   **Zhulin Tao**[3‡]   **Xun Yang**[1,4‡]

[1]University of Science and Technology of China    [2]Li Auto    [3]Communication University of China
[4]MoE Key Laboratory of Brain-inspired Intelligent Perception and Cognition, USTC

haotianling@mail.ustc.edu.cn, xyang21@ustc.edu.cn
{chenzequn, didonglin, mayongjia, lihao43, chenwei10}@lixiang.com
chenqy@cuc.edu.cn, taozl@cuc.edu.cn

## Abstract

Consistent pose-driven character animation has achieved remarkable progress in single-character scenarios. However, extending these advances to multi-character settings is non-trivial, especially when position swap is involved. Beyond mere scaling, the core challenge lies in enforcing correct Identity Correspondence (IC) between characters in reference and generated frames. To address this, we introduce EverybodyDance, a systematic solution targeting IC correctness in multi-character animation. EverybodyDance is built around the **Identity Matching Graph** (IMG), which models characters in the generated and reference frames as two node sets in a weighted complete bipartite graph. Edge weights, computed via our proposed Mask–Query Attention (MQA), quantify the affinity between each pair of characters. Our key insight is to formalize IC correctness as a graph structural metric and to optimize it during training. We also propose a series of targeted strategies tailored for multi-character animation, including identity-embedded guidance, a multi-scale matching strategy, and pre-classified sampling, which work synergistically. Finally, to evaluate IC performance, we curate the **Identity Correspondence Evaluation** benchmark, dedicated to multi-character IC correctness. Extensive experiments demonstrate that EverybodyDance substantially outperforms state-of-the-art baselines in both IC and visual fidelity.

## 1   Introduction

Character animation aims to generate video sequences from still images guided by specific pose sequences [1; 2]. Unlike text-driven generation focusing mainly on high-level semantic alignment [3; 4], it requires a dual fidelity: maintaining consistent visual appearance—including fine-grained details and accurately performing complex motion sequences [5; 6]. This requirement has generated significant research interest [7; 8; 9; 10; 11; 12].

Despite significant advances in single character animation generation (e.g. [8; 9; 13; 14; 12]), extending these methods to multi-character scenarios introduces unique challenges (see Figure 1). The key challenges are twofold. First, in multi-character scenarios, characters can swap relative

---

*Work done during an internship at Li Auto.

†Project Leader.

‡Corresponding authors.

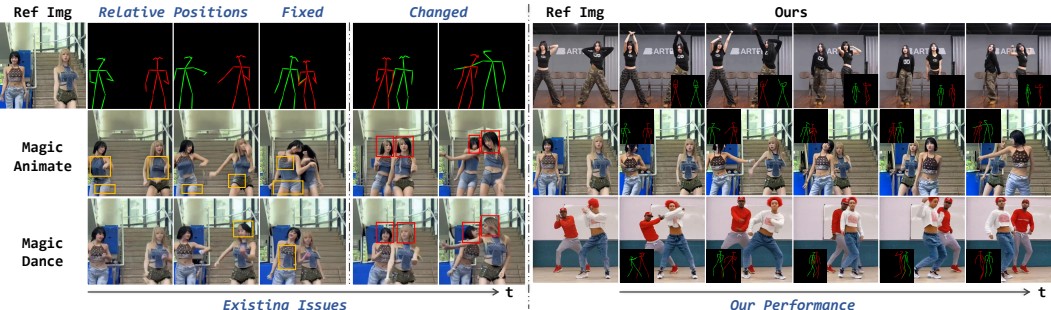

Figure 1: The left panel highlights the challenges of extending existing methods to multi-character scenarios. The yellow box indicates feature interference between characters, while the red box marks identity mismatches. The right panel illustrates our method's accurate identity correspondence.

positions, leading to identity confusion. Furthermore, the appearances of different characters can interfere with one another. Existing single-character animation approaches [14; 13; 9; 10; 12] are mainly based on implicit data-driven paradigms. In multi-character scenarios, such paradigm struggles to guarantee accurate one-to-one correspondence between generated and reference characters (see Section 4.3). Empirical results demonstrate that state-of-the-art methods often struggle to achieve satisfactory Identity Correspondence (IC) under these conditions (see Section 4.2).

To address these limitations, we propose an explicit modeling framework, which directly captures the correspondence between generated characters and their reference counterparts. Our method enforces correct IC between characters during training. Specifically, we introduce a weighted complete bipartite graph, **Identity Matching Graph** (IMG), whose two node sets represent generated/reference characters. Edge weights, derived by our proposed **Mask–Query Attention** (MQA), quantify the affinity between each generated/reference pair. A global matching score derived from the IMG provides a direct, optimizable objective for IC correctness. Integrating IMG into training achieves disentanglement of multiple characters and yields more accurate IC for multi-character animations. The construction process of IMG is **dynamic**, making it scalable for any number of characters.

To resolve the ambiguity of motion guidance in multi-character scenarios, we designed **Identity Embedded Guidance** (IEG). IEG provides clear anchors for each character throughout both training and inference. During the training phase, IEG and IMG work synergistically to create a guidance-supervision loop. To further strengthen the robustness of IC, we employ a suite of targeted improvements. First, to enforce the correct correspondence across the entire feature hierarchy, we introduce a multi-scale matching strategy. In addition, to address the long-tail distribution of the multi-character dataset, we propose a pre-classified sampling strategy to ensure that difficult and infrequent position-swap samples receive sufficient emphasis during training. To rigorously evaluate IC performance under complex multi-character conditions, we also present the Identity Correspondence Evaluation (ICE) benchmark, designed to challenge and compare SOTA methods on their ability to maintain correct IC.

Our main contributions are summarized as follows: *(1) Graph-Based IC Modeling*: We propose the Identity Matching Graph (IMG), a weighted complete bipartite graph that explicitly models IC in multi-character animation, whose edge weights are computed via our proposed Mask–Query Attention (MQA). *(2) Targeted Strategies*: We propose a series of targeted strategies, including identity-embedded pose guidance, a multi-scale matching strategy and a pre-classified sampling strategy, all tailored to multi-character animation. *(3) ICE Benchmark*: We curate the benchmark, ICE, for comprehensive evaluation in multi-character animation. Extensive evaluations demonstrate that our approach significantly outperforms SOTA baselines in both IC accuracy and visual fidelity.

## 2 Related Work

### 2.1 Diffusion Models

Diffusion Models gradually corrupt data by adding Gaussian noise and learn a reverse denoising process to model complex distributions [15; 16; 17; 18]. At inference, samples are generated by

starting from pure noise and iteratively denoising with the trained model [19; 20]. Extensions to latent diffusion operate in compressed feature spaces for efficient high-resolution and text-to-image generation [21]. Beyond images, diffusion frameworks produce temporally coherent videos for facial expression and dance generation [22; 9; 11]. Conditional diffusion enables flexible generation by guiding the reverse process with model-free or classifier-free cues [20; 23].

## 2.2 Video Generation

Early video synthesis relied on GAN-based [24] frameworks (e.g., TGAN [25]), which introduced temporal shift modules to enforce frame-to-frame coherence. Diffusion-based approaches [26] extend image diffusion models by integrating spatio-temporal conditioning or specialized temporal attention layers, as seen in Tune-A-Video's[27] tailored spatio-temporal attention and MagicVideo's [28] directed temporal attention module in latent space. Transformer-centric models such as Video Diffusion Transformer (VDT) [29] and Matten [30] leverage modular temporal and spatial attention (e.g., Mamba-Attention [31]) to capture long-range dependencies and global video context. Training-free extensions such as FreeLong [32] employ a SpectralBlend temporal attention mechanism to adapt pretrained short-clip diffusion models for long-video generation, maintaining both global consistency and local detail without additional training. Recent video super-resolution and editing techniques employ temporal-consistent diffusion priors to reduce flicker and preserve object appearance, further enhancing smoothness in tasks from animation to real-world scene synthesis [33].

## 2.3 Human Image Animation

Early GAN-based methods [34; 6; 35; 36; 37; 38; 39; 40] used appearance flow for feature warping but suffered from adversarial training issues such as mode collapse and motion inaccuracy [9]. More recent work [12; 9; 10; 7; 8; 11; 13; 41; 42; 43; 44; 45; 46] based on diffusion models, which offers stable training [47]. Disco [8] uses ControlNet [48] for disentangled pose–foreground–background control. ReferenceNet [9] improves fine-detailed consistency by injecting the appearance of a reference frame into the denoising UNet. Recent work has also made valuable contributions to multi-character scenarios. Ingredients [49] focuses on text-controlled multi-character layout, Follow-Your-Pose-V2 [50] focuses on scenes where characters maintain fixed relative positions.

## 3 Method

Identity Correspondence (IC), a one-to-one matching between each generated character and its counterpart in the reference frame, becomes especially critical when characters swap positions. Existing character animation methods [8; 9; 11; 13] typically rely on end-to-end training losses that capture only global similarity, often failing to enforce correct IC in such scenarios (see Section 4.3).

Section 3.1 introduces our formulation of the Identity Matching Graph (IMG). Section 3.2 explains how the IMG is constructed. Section 3.3 presents our targeted strategies for multi-character animation.

### 3.1 Problem Formulation

Concretely, we construct a weighted complete bipartite graph between the reference (ref) and the generated (gen) characters in each frame. The node set $\mathcal{R} = \{r_1, \ldots, r_m\}$ represents $m$ characters ordered from left to right in the reference frame (numbered 1 to $m$), while the node set $\mathcal{G} = \{g_1, \ldots, g_n\}$ with $n \leq m$ describes $n$ characters in the generated frame. We define the **Identity Matching Graph** (IMG) as the following bipartite graph:

$$\mathcal{B}_{\text{ID}} = (\mathcal{R}, \mathcal{G}, E, w), \quad E = \{(r_i, g_j) \mid 1 \leq i \leq m, \ 1 \leq j \leq n\}, \tag{1}$$

where the edge weight $w(r_i, g_j) \geq 0$ represents the affinity (potential correspondence) between $r_i$ and $g_j$ (see left panel of the Figure 2). ***During training***, for each generated frame we construct its IMG, yielding the set $\hat{E}$ of $n \times m$ edges (i.e., all possible correspondences between characters). We denote the edge set of $n$ ground-truth correspondences by $\mathcal{M}^*$. Since the edges in $\mathcal{M}^*$ represent the correct IC, our objective is to increase their weights by training the UNet [51]. Therefore, we use the following ratio $\mathcal{C}$ to quantify the correctness of IC as:

$$\mathcal{C} = \frac{\sum_{(r_i, g_j) \in \mathcal{M}^*} w(r_i, g_j)}{\sum_{(r_i, g_j) \in \hat{E}} w(r_i, g_j)} \ \in \ [0, 1], \mathcal{M}^* \subseteq \hat{E}. \tag{2}$$

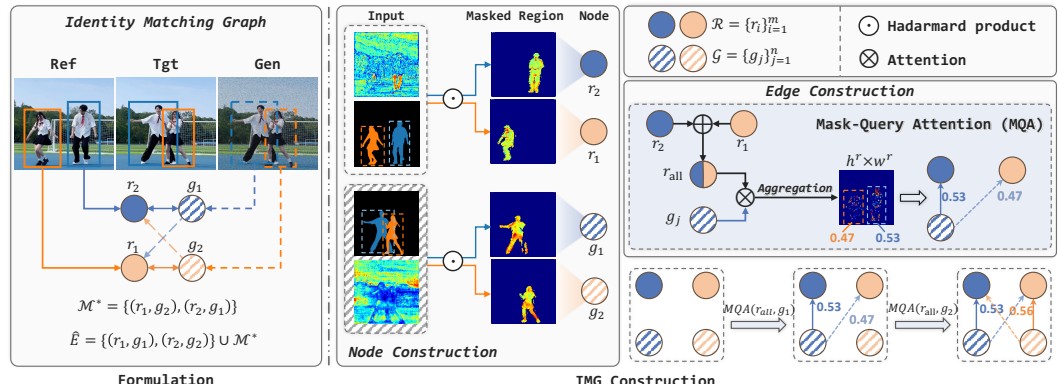

Figure 2: The left panel illustrates *what* is the IMG. The target (tgt) frame indicates the ground truth correspondence, indicating which edges belong to the set $\mathcal{M}^*$. The right panel shows *how* we build the IMG. Since the regions in $\mathcal{R}$ do not overlap spatially, we sum all $\{r_i\}_{i=1}^m$ representations into $r_{\text{all}}$.

Lower $\mathcal{C}$ indicates a more severe ambiguity. Optimizing $\mathcal{C}$ will force the model to learn correct IC, which will serve as a loss term during the training of the diffusion model. ***For example***, in the left panel of Figure 2, if the left generated character $g_1$ has a higher affinity to reference character $r_1$ than its true counterpart $r_2$, by the IMG construction in Section 3.2, the edge weight for $(r_2, g_1)$ will be a low value. Under the total loss defined in Equation 8, this low-weighted pairing incurs a penalty, thereby driving the model to learn the correct inter-character correspondence.

## 3.2 Identity Matching Graph Construction

***Node Construction.*** During training, the reference and generated frames are encoded into the latent space [47]. Therefore, we propose to use the corresponding masked regions in the latent space to represent each character, and then build the IMG nodes from those regions. We denote $\{M_i^{\text{r}}\}_{i=1}^m$ as the instance segmentation [52] masks of the $m$ reference characters, and $\{M_j^{\text{g}}\}_{j=1}^n$ represents the masks of the $n$ $(n \leq m)$ generated characters. Since instance segmentation is performed offline on the training set, we extract the masks for the generated frames from their corresponding ground-truth target frames. This approach circumvents potential drawbacks in both efficiency and accuracy.

In a chosen UNet [51] layer $\ell$, we extract the intermediate reference and generated feature, denoted as $\mathbf{f}^{\text{r}} \in \mathbb{R}^{c \times h^{\text{r}} \times w^{\text{r}}}$, $\mathbf{f}^{\text{g}} \in \mathbb{R}^{c \times h^{\text{g}} \times w^{\text{g}}}$, respectively. $(h^{\text{r}}, w^{\text{r}})$ and $(h^{\text{g}}, w^{\text{g}})$ are the sizes of the reference and generated latent maps. Each mask is interpolated to match the spatial resolution of the corresponding latent feature map: $\widetilde{M_i^{\text{r}}} = \text{Interp}(M_i^{\text{r}}) \in \{0,1\}^{h^{\text{r}} \times w^{\text{r}}}$, $\widetilde{M_j^{\text{g}}} = \text{Interp}(M_j^{\text{g}}) \in \{0,1\}^{h^{\text{g}} \times w^{\text{g}}}$. The latent regions corresponding to these segmentation masks are treated as graph nodes $r_i, g_j$ as:

$$r_i = \mathbf{f}^{\text{r}} \odot \widetilde{M_i^{\text{r}}}, \quad g_j = \mathbf{f}^{\text{g}} \odot \widetilde{M_j^{\text{g}}}, \tag{3}$$

$\odot$ denotes the Hadamard product, yielding reference nodes $\{r_i\}^{1:m}$ and generated nodes $\{g_j\}^{1:n}$.

***Edge Construction.*** To compute the edge weight $w(r_i, g_j)$ of the IMG, we propose the **Mask–Query Attention** to estimate the affinity between $g_j$ and $r_i$. It exploits the ability of the attention mechanism's [53; 9] to capture spatial dependence. Each generated character $\{g_j\}_{j=1}^{1:n}$ is transformed into a query matrix $Q_j \in \mathbb{R}^{(h^{\text{g}} \cdot w^{\text{g}}) \times d}$, and transfer each reference character $\{r_i\}_{i=1}^m$ to a key $K_i \in \mathbb{R}^{(h^{\text{r}} \cdot w^{\text{r}}) \times d}$. The attention map $A_i^j$ between $r_i$ and $g_j$ is calculated as:

$$A_i^j = \text{softmax}\left(\frac{Q_j K_i^{\top}}{\sqrt{d}}\right) \in \mathbb{R}^{(h^{\text{g}} \cdot w^{\text{g}}) \times (h^{\text{r}} \cdot w^{\text{r}})}. \tag{4}$$

$A_i^j[p, q]$ denotes the dependence from the $p$-th patch of $g_j$ to the $q$-th patch of $r_i$. We use the score $S_i^j$ to reflect the affinity between generated character $g_j$ and reference character $r_i$:

$$S_i^j = \sum_{q \in \mathcal{V}_i^{\text{r}}} \sum_{p \in \mathcal{V}_j^{\text{g}}} A_i^j[p, q], \quad \mathcal{V}_j^{\text{g}} = \{p \mid \widetilde{M_j^{\text{g}}}[p] = 1\}, \quad \mathcal{V}_i^{\text{r}} = \{q \mid \widetilde{M_i^{\text{r}}}[q] = 1\} \tag{5}$$

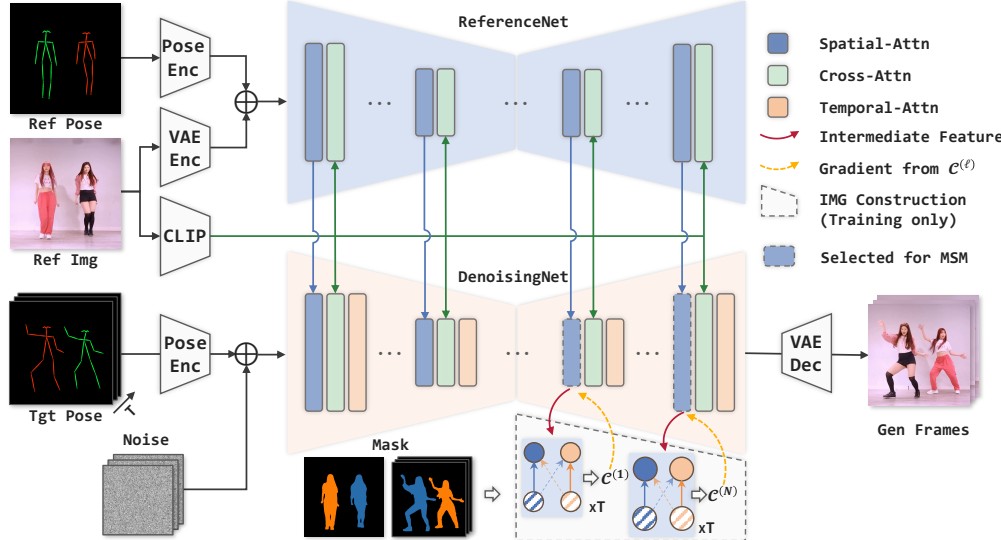

Figure 3: Training pipeline of EverybodyDance. We only construct the IMG during training. We additionally input the IEG of the reference image. ReferenceNet binds character identity by fusing the reference image's appearance to create identity-aware features that guide the DenoisingNet.

Equation 5 aggregates patch-level dependency $A_i^j$ into a node-level affinity score $S_i^j$, which reflects the overall correlation between the generated character $g_j$ and the reference character $r_i$. In the attention map $A_i^j$, each row corresponds to a patch in the generated latent $g_j$, and each column corresponds to a patch in the reference latent $r_i$. To quantify the relative affinity between $g_j$ and each reference character $\{r_i\}_{i=1}^m$, we first compute the set of affinity scores $\{S_i^j\}_{i=1}^m$. Each score $S_i^j$ aggregates patch-level attention scores from $A_i^j$ across the masked regions defined by $\widetilde{M}_j^{\text{g}}$ and $\widetilde{M}_i^{\text{r}}$. We then normalize these scores to obtain the final edge weights $\{w(r_i, g_j)\}_{i=1}^m$:

$$w(r_i, g_j) = \frac{S_i^j}{\sum_{i=1}^m S_i^j + \gamma} \in [0, 1), \quad \gamma = 10^{-8}. \tag{6}$$

Computing the pair-wise attention $A_i^j$ for all $m \times n$ pairs is inefficient. As shown in Figure 2, we aggregate all reference nodes into a single representation, $r_{all} = \sum_{i=1}^m r_i$. Each generated node $g_j$ then computes attention against $r_{all}$. This optimization reduces the computational complexity from $\mathcal{O}(m \cdot n)$ to $\mathcal{O}(n)$. Since $m$ and $n$ are dynamically determined by the number of segmentation masks, the IMG is built in a fully **dynamic** process that can be extended to any number of characters.

### 3.3 Targeted Strategies

***Identity-Embedded Guidance*** Existing methods rely solely on pose guidance without incorporating explicit identity information [54; 55; 35; 56], which complicates multi-character animation by lacking reliable identity cues for correct Identity Correspondence (IC). To resolve this, we introduce Identity-Embedded Guidance (IEG), which embeds identity into DWPose [54] by color-coding each skeleton (see Appendix for details). The IEG from each reference frame is also injected into its feature space.

These colored skeletons serve to mark reference characters and guide the placement of corresponding characters during generation, thereby enabling the model to differentiate identities during training and inference. We provide explicit input cues (IEG) and a matching loss (IMG). Both originate from the same segmentation masks to ensure alignment between guidance and supervision.

***Multi-Scale Matching*** To improve training robustness and ensure correct IC across different feature spaces, we perform Multi-Scale Matching (MSM) at $N$ selected UNet [51] layers $\{\ell\}_{\ell=1}^N$. At each

Table 1: Quantitative comparison on the ICE benchmark. Arrows indicate optimal direction (↓=lower better, ↑=higher better). AnimateAnyone* denotes the model fine-tuned on our dataset.

| Method | Frame Quality | | | | | Video Quality | |
| --- | --- | --- | --- | --- | --- | --- | --- |
| | SSIM↑ | PSNR*↑ | LPIPS↓ | L1↓ | FID↓ | FID-VID↓ | FVD↓ |
| AnimateAnyone [9] | 0.616 | 14.97 | 0.339 | 5.16E-05 | 59.19 | 32.057 | 364.85 |
| AnimateAnyone* [9] | 0.596 | 14.67 | 0.342 | 5.35E-05 | 54.71 | 31.274 | 358.31 |
| MimicMotion [14] | 0.621 | 15.00 | 0.338 | 5.48E-05 | 60.77 | 26.490 | 381.69 |
| MagicDance [12] | 0.508 | 13.81 | 0.424 | 1.33E-04 | 53.30 | 47.127 | 471.71 |
| MagicAnimate [10] | 0.614 | 13.95 | 0.369 | 6.36E-05 | 76.28 | 42.257 | 521.67 |
| UniAnimate [13] | 0.623 | 15.66 | 0.328 | 3.41E-05 | 44.38 | 26.696 | 295.56 |
| **EverybodyDance** | **0.654** | **16.93** | **0.304** | **2.86E-05** | **40.19** | **23.584** | **225.06** |

layer $\ell$ we construct its IMG $\mathcal{B}_{\mathrm{ID}}^{(\ell)}$ and compute layer-level IC score according to Equation 2:

$$\mathcal{C}^{(\ell)} = \frac{\sum_{(r_i,g_j)\in\mathcal{M}^*} w^{(\ell)}(r_i,g_j)}{\sum_{(r_i,g_j)\in\hat{\mathcal{E}}^{(\ell)}} w^{(\ell)}(r_i,g_j)} \ \in\ [0,1], \tag{7}$$

where $w^{(\ell)}(r_i, g_j)$ are the edge weights at layer $\ell$, $\mathcal{M}^*$ are the ground-truth correspondences, and $\hat{\mathcal{E}}^{(\ell)}$ is the full bipartite edge set at that layer. We then use the average of $\{-\mathcal{C}^\ell\}^{1:N}$ over all $N$ layers to be the matching loss $\mathcal{L}_{\mathrm{match}}$. Minimizing $\mathcal{L}_{\mathrm{match}}$ thus encourages the model to maximize IC correctness across all chosen scales, yielding more robust multi-character generation with accurate IC. The full pipeline is illustrated in Figure 3. The final training objective consists of standard diffusion reconstruction loss $\mathcal{L}_{\mathrm{diff}}$ and $\mathcal{L}_{\mathrm{match}}$, denoted as:

$$\mathcal{L} = \mathcal{L}_{\mathrm{diff}} \ + \ \lambda \mathcal{L}_{\mathrm{match}}, \quad \lambda > 0, \tag{8}$$

where $\lambda$ balances the frame quality against IC correctness.

***Pre-Classified Sampling*** Existing methods [11; 9; 10] typically select reference–target frame pairs randomly from a training video. However, in multi-character scenarios, challenging sample pairs, such as those involving position swaps, are relatively rare. To address this, we extract the position of each character. Then, with probability $\rho$ we draw from the pre-classified challenging swap pairs, and with probability $1 - \rho$ we conduct random sampling.

## 4 Experiment

### 4.1 Settings

**Quantitative Metrics.** To quantitatively evaluate the performance of different methods, we employ several widely used metrics, including L1 [57], PSNR* [58; 6], SSIM [59], LPIPS [60], FID [24], FID-VID [24], and FVD [61]. These metrics jointly provide a comprehensive evaluation.

**Baselines.** To validate the superiority of our method, we conduct extensive comparisons against several SOTA methods: MagicAnimate [10], AnimateAnyone [9], MagicPose [12], MimicMotion [14], Follow-Your-Pose-V2 [50] and UniAnimate [13].

**Dataset and Other Details.** We curated a custom multi-character dataset comprising approximately 800 video clips. For IC correctness evaluation, we introduce the ICE-bench, which contains 3,200 video frames. Our model is fine-tuned based on the AnimateAnyone framework using this dataset. For full descriptions of the training dataset and ICE-bench, other experiments, please refer to the Appendix.

### 4.2 Comparison Study

We evaluate our method, **EverybodyDance**, on the ICE-Bench using both quantitative metrics and qualitative showcases. As reported in Table 1, EverybodyDance achieves substantial improvements over its backbone model, AnimateAnyone: it reduces the FVD score by **38.3%**, indicating significantly improved video fidelity. To ensure these gains stem from our proposed targeted enhancements

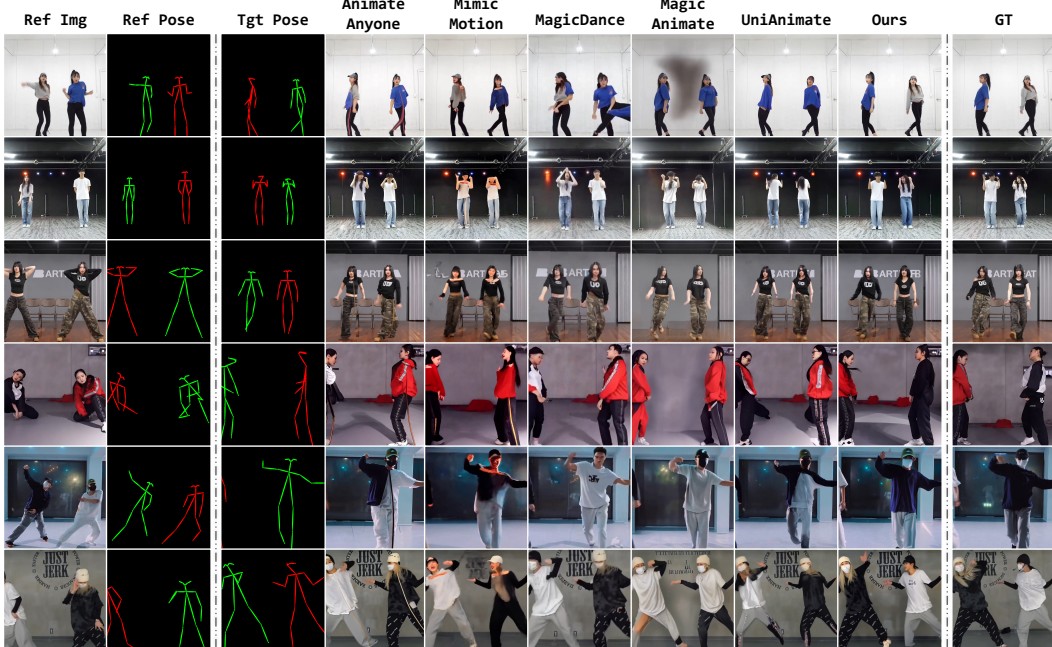

Figure 4: We compare our method with several state-of-the-art baselines. The last three rows illustrate three particularly challenging scenarios: (1) reference images exhibiting complex, non-standard poses; (2) target poses involving fewer character than the corresponding reference images; and (3) reference characters undergoing severe occlusion. Under these difficult conditions, our method consistently outperforms existing approaches, demonstrating accurate IC.

rather than dataset-specific biases, we additionally fine-tuned AnimateAnyone on our dataset; even so, it still fails to match the performance of EverybodyDance. Qualitatively, as shown in Figure 4 our approach consistently achieves accurate character identity correspondences in challenging scenarios such as position swaps, where existing methods often produce identity confusion or mismatches.

## 4.3 Ablation Study

To elucidate how our method enforces correct IC, we conduct a series of ablation experiments, summarized in Table 2. The experiments are categorized into the following three groups:

***Effectiveness of the Identity Matching Graph*** We compare our IMG-based approach against several ablation variants: *1)* t/w IEG: We fine-tune the backbone model using IEG rather than DWPose. *2)* End2End: We provide the IEG of reference image, allowing the model to learn IC in an end-to-end scheme. *3)* End2End-M: We further use masks over each character's region to enforce the model to focus on the corresponding region (see details in the Appendix).

As shown in the first group, introducing IEG alone yields some gains, while embedding identity cues (End2End and End2End-M) into the reference image's feature space enables partial performance improvements but remains insufficient. Only when IMG is incorporated to explicitly supervise character-to-character correspondence, the model achieves a dramatic improvement.

We also present qualitative comparison results in Figure 5. Figure 6 visualizes attention maps for both the IMG-based and end-to-end paradigms. We present visualizations of $r_{\mathrm{all}}$ alongside each generated character $g_j$ in Section 3.2. For each $g_1$ and $g_2$, arranged from left to right, we display its affinity scores with all reference characters. To enable direct comparison, we include the corresponding attention map visualizations from the End2End-M model. In the table, the columns labeled *IMG-$g_j$* and *End2End-M-$g_j$* respectively illustrate the attention maps of $g_1$ and $g_2$ over the reference image.

***Effectiveness of Multi-Scale Matching*** As demonstrated in the second group of experiments, a progressive increase in the number of matching layers leads to consistent improvements in overall performance.

Table 2: To facilitate analysis, the table is divided into three groups. MSM-$N$ refers to building the IMG using the last $N$ layers of the UNet, while PCS-$\rho$ denotes selecting pre-classified hard samples with a sampling ratio of $\rho$. In the *Full* setting, we set $N = 5$ and $\rho = 0.3$.

| Experiment Group | Experiment Settings | Frame Quality | | | | | Video Quality | |
|---|---|---|---|---|---|---|---|---|
| | | SSIM↑ | PSNR*↑ | LPIPS↓ | L1↓ | FID↓ | FID-VID↓ | FVD↓ |
| *IMG Effectiveness* | Full | 0.654 | 16.93 | 0.304 | 2.86E-05 | 40.19 | 23.584 | 225.06 |
| | Finetune | 0.596 | 14.67 | 0.342 | 5.35E-05 | 54.71 | 31.274 | 358.31 |
| | t/w IEG | 0.615 | 15.42 | 0.340 | 3.85E-05 | 48.78 | 29.804 | 319.96 |
| | End2End-M | 0.634 | 15.84 | 0.338 | 3.32E-05 | 45.05 | 28.464 | 285.09 |
| | End2End | 0.630 | 15.74 | 0.337 | 3.41E-05 | 45.23 | 28.789 | 289.69 |
| *MSM Settings* | MSM-4 | 0.649 | 16.81 | 0.308 | 2.96E-05 | 40.54 | 23.704 | 232.59 |
| | MSM-3 | 0.644 | 16.80 | 0.313 | 2.98E-05 | 40.49 | 24.566 | 232.47 |
| | MSM-2 | 0.641 | 16.68 | 0.317 | 3.02E-05 | 42.58 | 24.935 | 236.10 |
| | w/o MSM | 0.637 | 16.50 | 0.321 | 3.14E-05 | 41.26 | 25.507 | 256.01 |
| *PCS Settings* | PCS-0.5 | 0.654 | 16.88 | 0.311 | 2.95E-05 | 40.56 | 24.603 | 228.19 |
| | PCS-0.4 | 0.650 | 16.89 | 0.312 | 2.99E-05 | 41.73 | 23.708 | 234.56 |
| | PCS-0.2 | 0.652 | 16.72 | 0.311 | 2.96E-05 | 40.99 | 24.072 | 226.89 |
| | PCS-0.1 | 0.654 | 16.94 | 0.309 | 2.95E-05 | 40.47 | 23.592 | 227.32 |
| | w/o PCS | 0.632 | 16.23 | 0.329 | 3.07E-05 | 43.79 | 25.146 | 252.33 |
| *λ Settings* | $\lambda$-0.05 | 0.637 | 16.59 | 0.322 | 3.01E-05 | 41.76 | 23.243 | 233.34 |
| | $\lambda$-0.10 | 0.653 | 16.80 | 0.309 | 2.99E-05 | 42.04 | 23.691 | 234.35 |
| | $\lambda$-0.15 | 0.649 | 16.77 | 0.308 | 2.89E-05 | 42.62 | 24.093 | 234.44 |
| | $\lambda$-0.20 | 0.654 | 16.93 | 0.304 | 2.86E-05 | 40.19 | 23.584 | 225.06 |
| | $\lambda$-0.25 | 0.653 | 16.71 | 0.316 | 2.98E-05 | 40.47 | 23.156 | 230.18 |

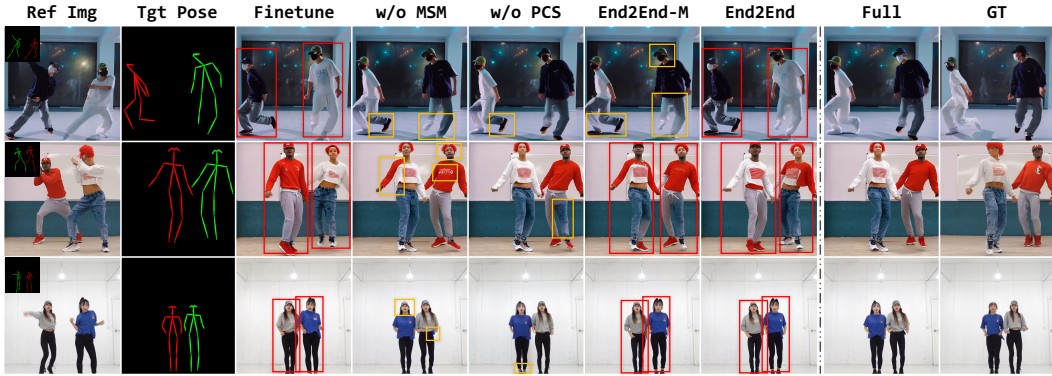

Figure 5: Qualitative comparison against different variants. Red boxes highlight cases of identity switch, while yellow boxes indicate instances of feature contamination.

**Effectiveness of Pre-Classified Sampling.** By comparing PCS under different sampling ratios, we find that a ratio of 0.3 achieves an optimal trade-off between hard sample abundance and diversity. This setting yields the best performance and effectively alleviates the long-tail data problem.

**Hyper-Parameters Analysis on the $\lambda$.** We conduct a sensitivity analysis on the hyper-parameter $\lambda$ introduced in Equation (8). Setting $\lambda$ to 0.20 achieves the best overall performance. This value offers an optimal trade-off between the diffusion reconstruction loss and the identity matching loss. Higher values cause the IC accuracy to plateau while slightly degrading the visual quality.

**Effectiveness of MQA.** We conduct an experiment in the Appendix to compare MQA with other similarity-based affinity calculation methods.

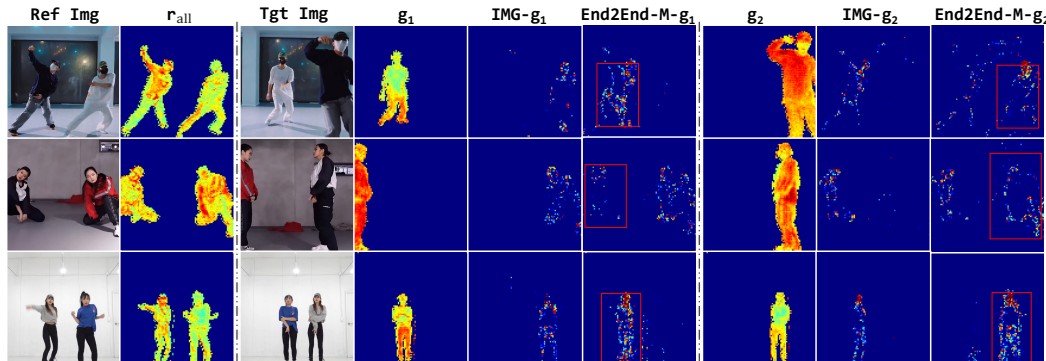

Figure 6: Ideally, the attention distribution should be primarily concentrated on the corresponding reference character located on the opposite side of $g_j$.

Table 3: Quantitative comparison on frame and video quality metrics. For more details about settings of this benchmark, please refer to [50].

| Method | Frame Quality | | | | | Video Quality | |
| | SSIM↑ | PSNR↑ | LPIPS↓ | L1↓ | FID↓ | FID-VID↓ | FVD↓ |
|---|---|---|---|---|---|---|---|
| DisCo [8] | 0.793 | 29.65 | 0.239 | 7.64E-05 | 77.61 | 104.57 | 1367.47 |
| MagicAnime [10] | 0.819 | 29.01 | 0.183 | 6.28E-05 | 40.02 | 19.42 | 223.82 |
| MagicPose [12] | 0.806 | 31.81 | 0.217 | 4.41E-05 | 31.06 | 30.95 | 312.65 |
| AnimateAnyone [9] | 0.795 | 31.44 | 0.213 | 5.02E-05 | 33.04 | 22.98 | 272.98 |
| Follow-Your-Pose-V2 [50] | 0.830 | 31.86 | 0.173 | 4.01E-05 | 26.95 | 14.56 | 142.76 |
| **EverybodyDance (Ours)** | **0.879** | **32.49** | **0.151** | **0.92E-05** | **26.01** | **12.68** | **127.36** |

## 4.4 Generalizability

***On Public Multi-character Benchmark.*** To validate the generalizability of our method, we also conducted comparisons with the publicly available benchmark provided by Follow-Your-Pose-V2 [50]. This benchmark is distinguished by frequent inter-person occlusions. As shown in Table 3, our method outperforms Follow-Your-Pose-V2 and other SOTA methods across all quality metrics. It should be noted that these metrics primarily reflect overall video fidelity. Since our method lacks explicit occlusion modeling, the foreground-background order during occlusions will be determined randomly.

***In Diverse Scenarios.*** We conducted a comprehensive quantitative evaluation to assess our method's capabilities in diverse scenarios. Specifically, we benchmarked its performance on challenging multi-character videos containing 3 to 5 individuals, and on the widely-used single-character TikTok [62] benchmark. As shown in Table 4, our proposed method demonstrates superior performance over all competing methods across all key metrics. For qualitative results, please refer to the Appendix.

***Cross-Video Motion Transfer.*** To assess the generalizability of our method for real-world applications, we conduct a cross-video motion transfer experiment. In this setting, a source video provides the motion template used to animate a diverse set of reference images. Moreover, we test the model's flexibility by reassigning character positions. We reorder the color-coded identities in the target IEG. The result, depicted in Figure 7, is a correctly rendered sequence where the characters' relative positions are swapped, underscoring our model's capacity for robust and flexible identity control.

## 5 Conclusion and Limitation

In this work, we introduce Everybody Dance, a framework that addresses the critical challenge of Identity Correspondence (IC) in multi-character animation. The core of our method is the Identity Matching Graph (IMG), which formalizes the ambiguous problem of IC correctness into an explicit, optimizable graph-structural metric. To construct this graph, our Mask-Query Attention (MQA) efficiently computes edge weights. This graph-based loss works in synergy with our Identity-Embedded Guidance (IEG) together, they form a cohesive guidance-supervision architecture. Finally,

Table 4: Quantitative comparison on multi-character and single-character benchmarks. We group metrics into Frame Quality and Video Quality. Best results are in **bold**.

| Scene | Method | Frame Quality | | | | Video Quality | |
|---|---|---|---|---|---|---|---|
| | | SSIM↑ | PSNR*↑ | LPIPS↓ | FID↓ | FID-VID↓ | FVD↓ |
| *More Character* | AnimateAnyone [9] | 0.606 | 14.70 | 0.370 | 56.66 | 35.356 | 401.51 |
| | AnimateAnyone* [9] | 0.607 | 14.92 | 0.363 | 64.34 | 36.308 | 406.08 |
| | UniAnimate [13] | 0.640 | 15.62 | 0.338 | 57.75 | 29.030 | 348.91 |
| | **Ours** | **0.671** | **16.68** | **0.315** | **42.84** | **23.571** | **261.01** |
| *Single Character* | AnimateAnyone [9] | 0.768 | 17.85 | 0.280 | 52.15 | 25.864 | 209.14 |
| | AnimateAnyone* [9] | 0.764 | 17.19 | 0.291 | 62.52 | 25.943 | 213.68 |
| | End2End | 0.770 | 17.65 | 0.288 | 45.25 | 22.515 | 186.34 |
| | **Ours** | **0.772** | **17.78** | **0.279** | **40.56** | **20.294** | **163.85** |

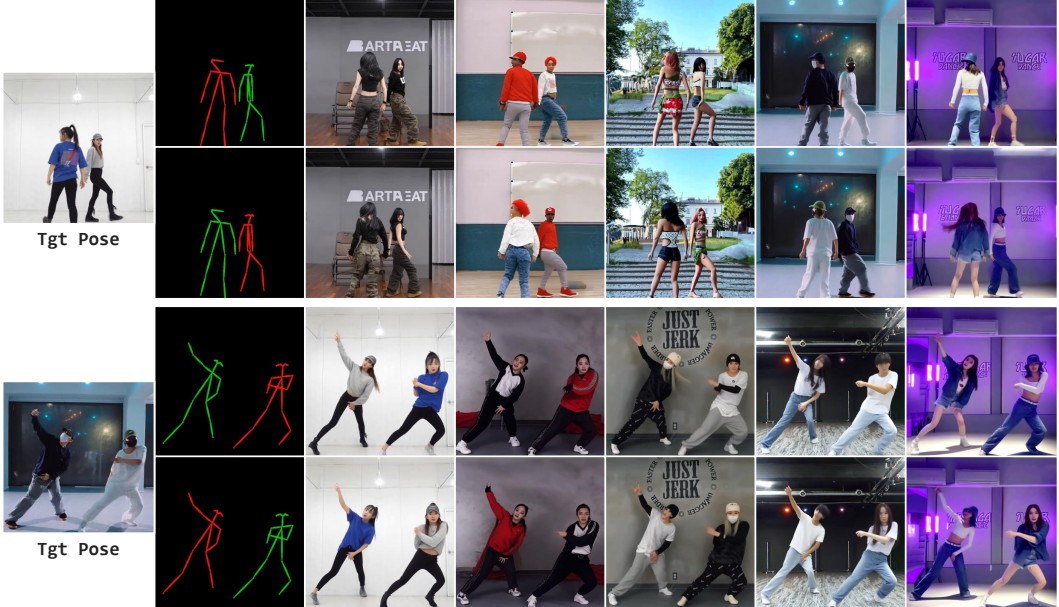

Figure 7: We employ a multi-person video clip as the source pose and use various references to generate animations. We also swap the relative positions of characters in the target pose.

we enhance IC robustness through two targeted strategies: Multi-Scale Matching (MSM) enforces correctness across multiple feature hierarchies, while Pre-Classified Sampling (PCS) addresses challenging, rare training samples. These contributions enable our model to significantly improve identity consistency and visual quality in complex multi-character scenes.

However, our current method is unable to effectively handle scenarios with severe inter-character occlusion. Incorporating 3D datasets [63; 64; 65] presents a promising direction for future work to address this. Furthermore, the performance of our method depends on the accuracy of the upstream instance segmentation model. Finally, our current quantitative evaluation still relies on proxy metrics that measure overall video fidelity. Designing dedicated metrics that can directly and quantitatively evaluate IC correctness remains a significant open problem.

# 6 Acknowledgement

The authors appreciate the generous support of Li Auto, which provided the financial backing and essential computational resources that made this research possible. The authors also thank our colleagues at the University of Science and Technology of China, Li Auto, and Communication University of China for their insightful discussions and support throughout this project.

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

# Appendix

## A    Preliminary: ReferenceNet Based Character Animation

Character animation aims to synthesize realistic character videos from a single reference image and a driving pose sequence. Formally, given a reference image $x_{\text{ref}}$ and a target pose sequence $\{p_t\}_{t=1}^{L}$, the goal is to generate a video $\{x_t\}_{t=1}^{L}$ where each frame $x_t$ maintains visual consistency with $x_{\text{ref}}$ while conforming to the pose $p_t$.

Most of character animation approach [12; 10; 9; 11; 13] builds upon the Stable Diffusion [47] framework, which performs denoising in the latent space. Let $Enc$ and $Dec$ denote the encoder and decoder of the latent diffusion model. The reference image is first encoded into a latent representation $z_{\text{ref}} = Enc(x_{\text{ref}})$, and each frame is generated from noisy latent inputs $z_T \sim \mathcal{N}(0, I)$ through a conditional denoising process:

$$z_0 = \text{Denoise}(z_T, \{p_t\}_{t=1}^{L}, x_{\text{ref}}), \tag{9}$$

where the denoising process is iteratively performed by a UNet-based network to recover $z_0$, which is then decoded by $Dec(z_0)$ to reconstruct the video frame.

To preserve the appearance consistency of $x_{\text{ref}}$, [9] introduce ReferenceNet, a UNet-like structure $R - \text{Net}$ designed to extract spatially detailed features from the reference image. Specifically, $R - \text{Net}$ produces intermediate features $f_{\text{ref}} \in \mathbb{R}^{H \times W \times C}$, which are fused into the main denoising UNet $\epsilon_\theta$ via a spatial-attention mechanism:

$$\text{Attn}_{\text{spatial}}(x_1, x_2) = \text{SelfAttention}\left(\text{Concat}(x_1, \text{Repeat}(x_2, t))\right). \tag{10}$$

Here, $x_1 \in \mathbb{R}^{t \times h \times w \times c}$ is the feature from the denoising UNet, and $x_2 \in \mathbb{R}^{h \times w \times c}$ from ReferenceNet. The operation repeats $x_2$ along the temporal axis and performs attention, then extracts the first half as the refined output, ensuring that spatial detail flows from the reference into each frame.

In addition, high-level semantic features from the CLIP image encoder are used in cross-attention to condition the denoising on global content, complementing ReferenceNet's local details.

By aligning both low-level and high-level cues from the reference image, and integrating them into the diffusion denoising pipeline, the ReferenceNet significantly enhances the temporal and spatial fidelity of character animations. Its design ensures efficient inference, as $R - \text{Net}$ is executed only once per sequence, while maintaining consistency across all video frames.

## B    Other Details

### B.1    Identity-Embedded Guidance

Although AlphaPose [55] provides built-in identity tracking alongside multi-character pose estimation, we observed that its tracking module is not sufficiently reliable for complex multi-character scenarios. In particular, we encountered the following common failure cases: (1) **Identity Confusion**: Different individuals are incorrectly assigned the same identity label due to visual similarity (e.g., similar clothing); (2) **Cross-identity Misassignment**: Skeletal data from multiple individuals are incorrectly associated, resulting in identity mismatches; (3) **Temporal Inconsistency**: The identity assigned to a person changes from frame to frame, often due to occlusion or tracking errors.

These identity-related failures can severely hinder the downstream learning of accurate identity correspondence by introducing noisy supervision and temporal jitter. To mitigate such tracking errors, we decouple pose estimation and identity assignment: multi-character pose maps are extracted using DWPose [54], while consistent identity anchors are derived from instance-level bounding boxes produced by SAM2 [52]. This hybrid approach yields pose embeddings that are structurally faithful and identity-discriminative, reducing errors caused by tracking failures.

We utilize DWPose to extract multi-character poses from individual image frames, where each detected pose is represented by a set of 2D keypoints with corresponding confidence scores. However, the output poses are unordered and lack explicit identity labels. To address this limitation, we use SAM2 to generate instance-level bounding boxes, with each bounding box assigned a unique identity label, which will serve as a persistent identity label between frames.

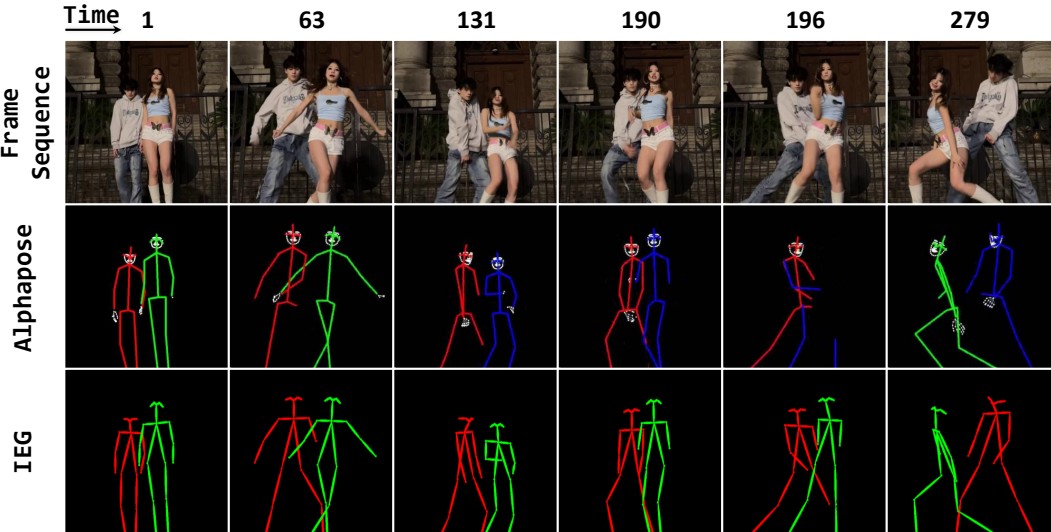

Figure 8: AlphaPose exhibits severe identity inconsistency. Using the skeleton color from frame 1 as identity reference, the same female character is mistakenly reassigned from green (frame 1) to blue (frame 131), and the male character from red to blue (frame 279). In contrast, our IEG maintains consistent identity assignment throughout the sequence.

For each candidate pose, we calculate the proportion of its keypoints that fall within each bounding box. A pose-box pairing is accepted only if the ratio of enclosed keypoints exceeds a predefined threshold, ensuring robustness in cluttered or partially occluded scenes. This strategy effectively preserves the temporal continuity of identity embeddings (see Figure 8).

Each matched skeleton is labeled with an identity-specific color to distinguish different characters. This color-encoded representation serves as our Identity-Embedded Guidance (IEG). This strategy obviates the need for additional training or hyperparameter tuning, exhibits strong robustness in complex multi-character scenarios involving position swaps, and retains high generalizability across different pose estimation backbones and instance-level segmentation frameworks.

## B.2 Training Pipeline

To formally revisit the core training pipeline described in the main text, we summarize the overall procedure as follows. During training, the model learns to generate multi-character video frames with correct IC by jointly optimizing the diffusion reconstruction loss and the identity matching graph (IMG)–based IC loss. Initially, the target frame sequence $\mathbf{x}_0$ is passed through a VAE encoder to obtain the latent representation $\mathbf{z}_0$, and instance segmentation masks $\{M_i^r\}_{i=1}^m$ are extracted via SAM [52]. At each diffusion timestep $t$, noise $\varepsilon$ is injected into $\mathbf{z}_0$ to produce $\mathbf{z}t$, which, alongside identity-embedded pose guidance $\mathbf{c}^r, \mathbf{c}^t$ and semantic features $\mathbf{s}^r$ of the reference image (according to [9]; we additionally include $\mathbf{c}^r$ to inject identity information), is processed by the UNet backbone to predict $\hat{\varepsilon}$. Concurrently, at $N$ selected UNet layers, the intermediate features $\mathbf{f}^r, \mathbf{f}^g$ and interpolated masks $\widetilde{M}^r, \widetilde{M}^g$ are used to construct IMG nodes $r_i, g_j$. Mask–Query Attention computes affinities $w^{(\ell)}(r_i, g_j)$, from which the layer-wise consistency score $\mathcal{C}^{(\ell)}$ is derived and aggregated into the matching loss $\mathcal{L}$match. With probability $\rho$, challenging swap-pair samples are selected via pre-classified sampling to emphasize identity-switch scenarios. As shown in the Algorithm 1.

## B.3 Inference Pipeline

During inference, EverybodyDance generates multi-character video frames conditioned on a reference frame, a reference pose, and a driving pose sequence, without constructing the IMG or computing any matching loss. The pose guidance is pre-processed in IEG $\mathbf{c}^r, \mathbf{c}^t$ to indicate character identities. The driving pose $\mathbf{c}^t$ is added with initial noise to predict $\hat{\varepsilon}$. The iterative denoising process follows the standard DDIM schedule, gradually refining $\mathbf{z}_t$ back to $\mathbf{z}_0$. Finally, the VAE decoder reconstructs

---

**Algorithm 1** Training Pipeline of EverybodyDance

---

**Require:** Target frame $\mathbf{x}_0$, IEG $\mathbf{c}^r$, $\mathbf{c}^t$, masks $\{M_i^r\}$, sampling ratio $\rho$, reference image, total training steps $S$.
**Ensure:** Model parameters $\theta$
 1: **for** step = 1 to $S$ **do**
 2:     Sample challenging pair w.p. $\rho$ else random
 3:     $\mathbf{z}_0 \leftarrow \text{VAE.Encode}(\mathbf{x}_0)$
 4:     Extract reference semantic features $\mathbf{s}^r$ by $R-\text{Net}$
 5:     $\varepsilon \sim \mathcal{N}(0, I)$, $\mathbf{z}_t \leftarrow \sqrt{\bar{\alpha}_t}\,\mathbf{z}_0 + \sqrt{1-\bar{\alpha}_t}\,\varepsilon$
 6:     $\hat{\varepsilon} \leftarrow \text{UNet}_\theta(\mathbf{z}_t, \mathbf{c}^r, \mathbf{c}^t, \mathbf{s}^r, t)$
 7:     $\mathcal{L}_{\text{diff}} = \|\varepsilon - \hat{\varepsilon}\|^2$
 8:     **for** $\ell = 1$ to $N$ **do**
 9:         Extract $\mathbf{f}^r, \mathbf{f}^g$, interpolate masks, construct nodes $\{r_i = \mathbf{f}^r \odot \widetilde{M}_i^r\}_{i=1}^m$.
10:         **for** $j = 1$ to $n$ **do**
11:             $g_j = \mathbf{f}^g \odot \widetilde{M}_j^g$, $r_{\text{all}} = \sum_{i=1}^m r_i$
12:             Compute $\{w^{(\ell)}(r_i, g_j)\}_{i=1}^m = \text{MQA}(r_{\text{all}}, g_j)$.
13:         **end for**
14:         Compute $\mathcal{C}^{(\ell)}$
15:     **end for**
16:     $\mathcal{L}_{\text{match}} = \frac{1}{N}\sum_\ell -\mathcal{C}^{(\ell)}$
17:     Update $\theta \leftarrow \theta - \eta \nabla_\theta(\mathcal{L}_{\text{diff}} + \lambda \mathcal{L}_{\text{match}})$
18: **end for**

---

---

**Algorithm 2** Inference Pipeline of EverybodyDance

---

**Require:** Reference image, driving poses IEG $\{\mathbf{c}^t\}_{t=1}^T$, reference IEG $\mathbf{c}^r$
**Ensure:** Generated frames $\{\hat{\mathbf{x}}_t\}_{t=1}^T$
 1: $\mathbf{z}_t \leftarrow \mathcal{N}(0, I)$
 2: Extract semantic features $\mathbf{s}^r$ by $R-\text{Net}$
 3: **for** $t = T$ to 1 **do**
 4:     Sample $\hat{\varepsilon} \leftarrow \text{UNet}_\theta(\mathbf{z}_t, \mathbf{c}^r, \mathbf{c}^t, \mathbf{s}^r, t)$
 5:     Compute $\mathbf{z}_{t-1}$ via DDIM [20]
 6: **end for**
 7: $\hat{\mathbf{x}}_t \leftarrow \text{VAE.Decode}(\mathbf{z}_0)$

---

the RGB frame, producing a temporally coherent multi-character video with accurate IC. As shown in the Algorithm 2.

### B.4   End2End & End2End-M

We introduce two alternative training strategies to assess the effectiveness of the Identity Matching Graph (IMG). End2End refers to a fully end-to-end training pipeline where the IMG construction is not involved, and the model is trained solely with standard video generation losses. End2End-M, on the other hand, replaces the original Identity Correspondence (IC) loss derived from the IMG with a simplified constraint: an L2 loss computed over the masked region of the reference image. This provides a coarse form of identity guidance without constructing the full identity matching graph. These two strategies represent training without identity constraints (End2End) and with weak identity constraints (End2End-M), respectively.

## C   ICE-Bench

Although Follow-Your-Pose-V2 [50] introduced the *Multi-Character* benchmark for multi-character animation, it features relatively simple character interactions and only limited occlusion scenarios. To address this critical gap, we introduce Identity Correspondence Evaluation benchmark (ICE-Bench), the first multi-character benchmark to evaluate IC performance in multi-character animation tasks.

ICE-Bench consists of carefully curated video clips dance video clips totaling more than 3,200 frames. ICE-Bench is deliberately designed to stress test identity correspondence in challenging interaction scenarios. We adopt a multi-criteria filtering strategy to ensure that the benchmark presents sufficient challenges. We first select clips that exhibit clear multi-character interactions, such as positional exchanges and occlusions, discarding those with minimal interaction. Second, we retain videos with stable lighting, minimal blur, and clear joint visibility to ensure visual quality and reliable pose extraction. Finally, we balance diversity and difficulty by including varied dance genres and environments.

## D  More Experiment Details

### D.1  Implementation Details

We employ the Animate Anyone [9] as the backbone for our animation pipeline. In our setup, both the ReferenceNet and DenoisingNet utilize a shared pose guider. The construction of the IMG does not involve adding any additional modules; we directly leverage the original spatial attention blocks already present in the DenoisingNet. The training process is divided into two stages. *Stage 1* focuses on single-frame spatial quality, while *Stage 2* prioritizes temporal coherence in video sequences. Stage 1 was trained for 5,000 steps while Stage 2 was trained for 1,000 steps, with the VAE [66] encoder and CLIP [67] encoder frozen throughout. In Stage 1, the ReferenceNet, DenoisingNet, and Pose Guider are trainable. A batch size of 32 is applied, using center-cropped 768×768 resolution images. For Stage 2, only the temporal attention modules are trainable, with a batch size of 8 and training sequences comprising 24 consecutive frames sampled at 3-frame intervals. Video frames are processed at 512×512 resolution. We set the learning rate at $2.0 \times e^{-5}$ and use the Adam [68] optimizer. During the inference phase, we employ the DDIM scheduler with 50 denoising steps. We set the classifier-free guidance [69] scale to 3.5.

### D.2  Dataset

We construct MultiDance dataset, a dataset tailored for multi-character animation. Existing datasets are predominantly designed for single-person motion transfer and lack both rich multi-character interaction. MultiDance comprises 814 multi-character dance video clips at 1080p resolution, totaling 257K frames. It covers a diverse range of challenging scenarios, including dynamic position exchange and partial occlusion. Notably, approximately 30% of the frames contain relatively complex interactions (e.g. positional swaps or occlusion). Dance styles span various categories such as pop dance, street dance, aerobics, and ballet, filmed in both indoor (well lit) and outdoor (evenly illuminated) environments.

## E  More Experiments

### E.1  Training Curve

We record the IC score (cooresponding to the matching loss $\mathcal{L}_{\text{match}}$) curve throughout the training process. As shown in Figure 10, which clearly reflects the model's progressively improving ability to capture identity correspondences — increasing from 0.55 to approximately 0.70.

### E.2  More Characters

To further evaluate the effectiveness of our method in more complex settings, we increase the number of characters involved in the generation process. To assess generalization capability, we employ a video clip as the source of the pose sequence and use a single reference image to guide the generation of an entire video. Importantly, the relative spatial arrangements of the characters differ between the reference image and the target poses, presenting a more challenging scenario for preserving accurate identity correspondences. The qualitative results are presented in Figure 11.

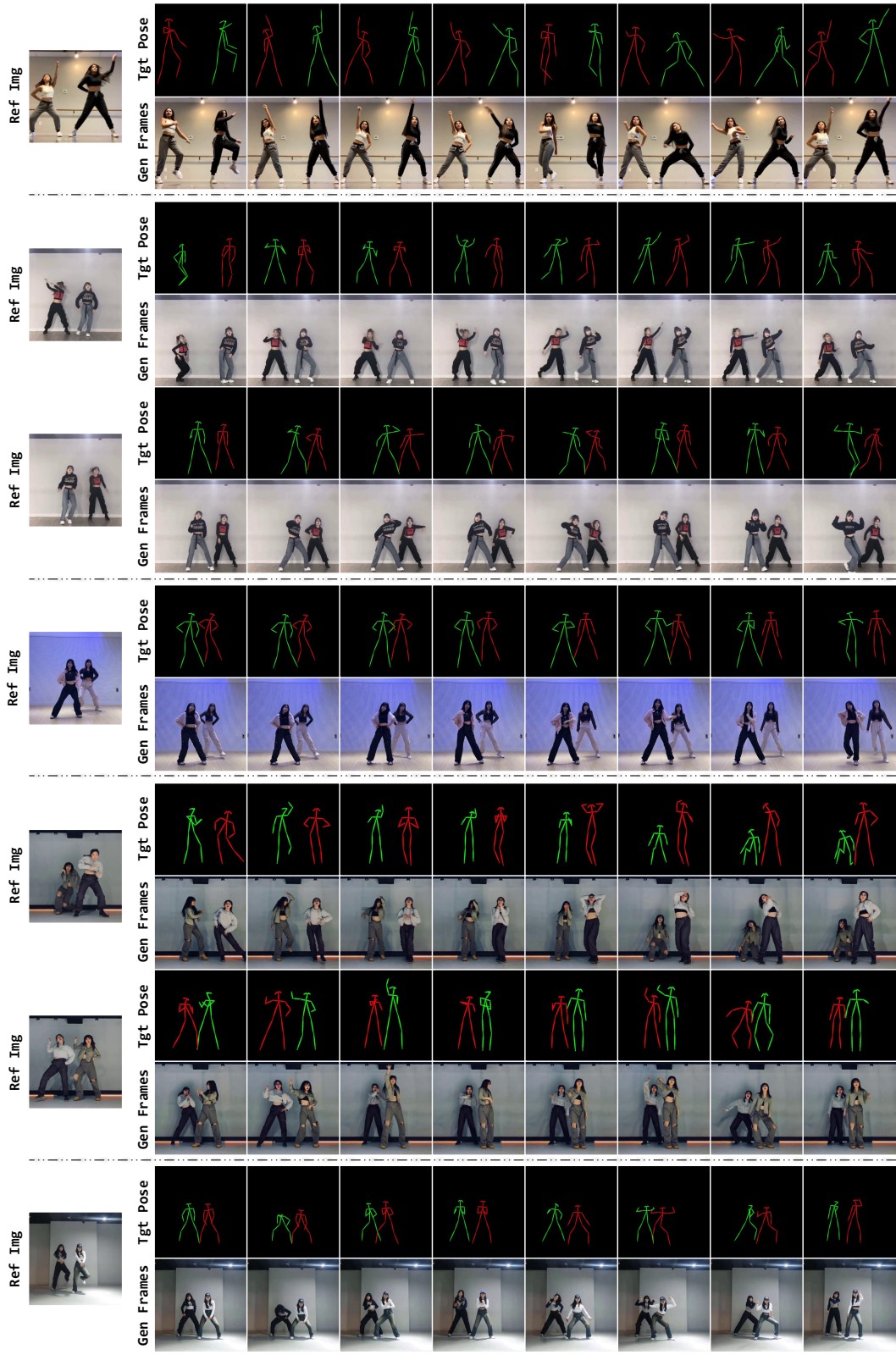

Figure 9: Our performance on Follow-Your-Pose-V2 *Multi-Character* pulic benchmark.

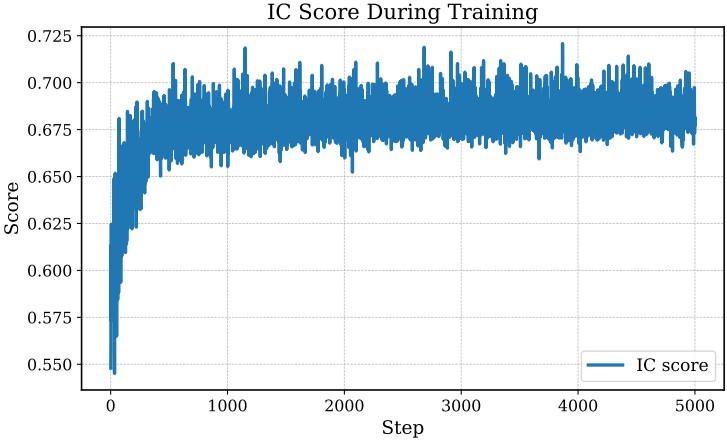

Figure 10: Identity correspondence score during stage 1 training.

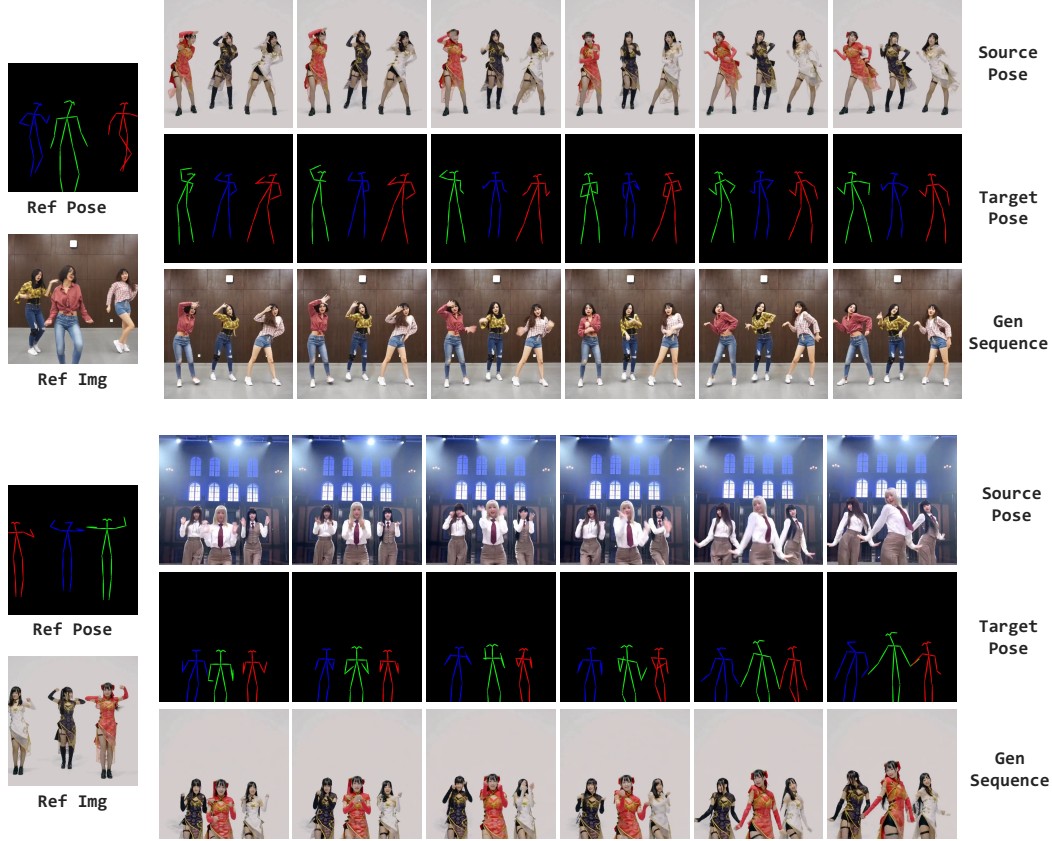

Figure 11: Even with more character identities, our method consistently maintains accurate identity correspondence.

Table 5: Quantitative comparison of different affinity computation methods on our ICE benchmark. Our proposed IMG+MQA approach demonstrates superior performance in both frame-level quality and temporal video consistency.

| Method | SSIM ↑ | PSNR* ↑ | LPIPS ↓ | FID ↓ | FID-VID ↓ | FVD ↓ |
|---|---|---|---|---|---|---|
| CrossAttn-Based | 0.648 | 15.69 | 0.315 | 45.82 | 29.22 | 306.75 |
| Sim-Based | 0.629 | 16.25 | 0.329 | 47.09 | 24.35 | 269.14 |
| **Ours (IMG+MQA)** | **0.654** | **16.93** | **0.304** | **40.19** | **23.58** | **225.06** |

## F  Further Analysis on Affinity Computation

While our primary approach utilizes the Identity Matching Graph (IMG) with Mask-Query Attention (MQA), we also explored alternative paradigms for affinity computation to further validate our design choices. One inspiring direction comes from methods like **Ingredients** [49], which use per-character CLIP embeddings for layout control. We designed and evaluated two alternative variants optimized with a standard cross-entropy classification loss. The implementation details are as follows:

- **CrossAttn-Based:** In this variant, we first segment the reference image and encode each of the $m$ characters into distinct identity features using a CLIP encoder. Within the Cross-Attention layers of our model, instead of attending to a single CLIP embedding for the entire image, we compute the affinity between the latent features of the generated frame and each of the $m$ individual character embeddings. After applying a softmax function and masking the background, this process yields an affinity map of shape $[HW, m]$, where each entry represents the probability of a given latent patch belonging to one of the reference characters.

- **Sim-Based:** This approach relies on direct feature similarity. We use segmentation masks to extract intermediate features for the $m$ reference characters and $n$ generated characters. We then compute an $[n, m]$ affinity matrix by taking the dot-product between each generated and reference character's feature representation, followed by a softmax operation. Each entry $(i, j)$ in this matrix indicates the similarity score between the $i$-th generated character and the $j$-th reference character.

As shown in Table 5, our proposed IMG+MQA framework outperforms both variants across all quantitative metrics. The `CrossAttn-Based` method, while reasonable, relies on an external CLIP space that is not inherently tied to the generative model's internal feature representation, creating a disconnect. The `Sim-Based` approach performs worse, particularly struggling to differentiate between characters with visually similar features. We chose the IMG+MQA paradigm because it computes affinity using the model's native spatial attention scores, which are intrinsically coupled with the denoising and generation process. This inherent coupling ensures that the supervision signal for identity correspondence directly influences the most relevant parts of the network responsible for spatial layout and appearance, leading to superior performance.

## G  User Study

To comprehensively evaluate the perceptual quality of the generated multi-character videos, we conduct a user study from three complementary perspectives: fidelity, coherence, and identity correspondence (IC) accuracy.

For the study, we collected approximately 200 sets of human feedback from a group of university-educated participants. In each evaluation session, participants were shown the video clips generated by all competing methods for the same input. They were then asked to score each video on a scale from 1 (worst) to 5 (best) for the three criteria independently. The final user preference scores are aggregated to provide a quantitative comparison, as shown in Table 6.

Specifically, fidelity measures how realistic and visually appealing each frame appears; coherence assesses the temporal consistency and motion smoothness across frames; and IC accuracy evaluates whether the identity correspondences between the generated video and the reference images are correctly maintained as expected.

Table 6: User study results based on three quality dimensions: Fidelity, Coherence, and Identity Correspondence (IC) Accuracy.

| Method | Fidelity | | Coherence | | IC Accuracy | |
|---|---|---|---|---|---|---|
| | Mean | Var. | Mean | Var. | Mean | Var. |
| **EverybodyDance** | **4.40** | **0.48** | **4.32** | **0.55** | **4.57** | **0.38** |
| UniAnimate | 3.01 | 1.22 | 3.61 | 1.09 | 2.12 | 0.80 |
| MagicDance | 3.73 | 1.20 | 2.80 | 1.34 | 1.99 | 0.87 |
| AnimateAnyone | 3.05 | 1.31 | 3.25 | 1.28 | 2.33 | 1.07 |
| MagicAnimate | 2.13 | 1.14 | 2.25 | 1.26 | 1.83 | 0.69 |

In the user study, participants are shown a series of video clips generated by different methods and are asked to score them on a scale from 1 (worst) to 5 (best) for each of the three criteria independently. The final user preference scores are then aggregated to provide a quantitative comparison across different methods, as shown in Table 6.

