# OpenReview forum: "EverybodyDance: Bipartite Graph–Based Identity Correspondence for Multi-Character Animation"
_NeurIPS.cc/2025/Conference — NeurIPS 2025 poster_

### Official Review · Reviewer_Mm46 · 2025-07-01

**Clarity:** 4
**Significance:** 3
**Originality:** 3
**Rating:** 5
**Confidence:** 4

**Summary:**

This paper proposes EverybodyDance, which aims to address the issue of identity consistency in pose-conditioned multi-person dance videos. Specifically, EverybodyDance adopts the backbone of AnimateAnyone. Its input includes a reference image of multiple humans (typically two or three people), a pose map (with different subjects marked in different colors), and target poses, while the output is a video sequence.

The highlight of this work lies in a simple yet reasonable training strategy designed to enhance the network's identity awareness. Put simply, it extracts the deep features of all subjects from both the ground truth and the generated frames using masks, and applies a matching loss that encourages low similarity between different identities and high similarity within the same identity. This idea (at least seemingly) is straightforward and practical, and effectively improves identity alignment.

The authors also collect a new multi-person (mainly two-person) dance dataset to train and validate their method.

Although the core idea of the paper is quite simple and straightforward, it shows significant improvements and could be highly beneficial to the field of multi-subject video generation.

**Questions:**

On the design of affinity computation, the authors take an equal and unique treatment to different kind of features into single "attention map" (Line 150). However, is there any possibility to treat different parts (e.g., hair, clothes, pants, decorations...) separately? Will such "equal and unique" way lose identity details?

**Ethical Concerns:**

["NO or VERY MINOR ethics concerns only"]

**Final Justification:**

The rebuttal generally resolve my questions. I keep the original score.

However, the reviewers DO need to revise details I mentioned (including data occlusions, using/discussing 3D interaction data...).

**Limitations:**

There is a short limitation. But I didn't see "potential negative social impact" of this work. The authors do need to append it further.

Meanwhile, this work includes a new dataset&benchmark. **Do the collected data involve copyright issues and privacy issues**? I am not sure about it.

**Paper Formatting Concerns:**

No.

**Quality:**

3

**Strengths And Weaknesses:**

**Strength**
+ The core idea is very simple, reasonable, and easy to extend.

+ The authors introduce a dual-person dance dataset (albeit a small one) and propose a benchmark based on it.

+ Experiments show that the proposed identity matching method significantly improves identity consistency. The ablation study is also fairly comprehensive.

+ The writing is clear and easy to understand.

**Weaknesses**

- In the authors’ demo, when occlusion occurs between subjects, the pose and generated video exhibit abrupt changes. Does this indicate that the authors intentionally avoided evaluating performance under occlusion? Does this imply that their method may not generalize well to scenarios involving strong interactions?

 - When discussing the paper’s limitations, the authors cite the lack of high-quality data. However, there exist many high-quality 3D human-human interaction datasets, such as You2Me (Ng et al., 2020), InterHuman (Liang et al., 2023), and DD100 (Siyao et al., 2024). The authors neither discuss whether training on such high-quality 2D/3D labeled data could improve performance, nor do they cite these works.

---

> ### Author Rebuttal · Authors · 2025-07-28
>
> ## **I. Concern about Potential Loss of Identity Details**
> ---
> Thank you for your insightful question concerning the granularity of our affinity computation. To answer this question, we first provide an analysis based on the training process, and subsequently design an experiment to verify it.
>
> ### *A. Principle Analysis*
> During training, the matching loss and the diffusion model’s L2 loss work together; therefore, the loss of identity details would be penalized by the L2 loss. As shown in Figure 1, our method accurately preserves the identity details such as hairstyles and shoes. For example, the woman in white and the man in orange. Your suggestion to treat different parts separately is very insightful and could be achieved with more advanced segmentation models, potentially improving the identity appearance details. Below, we design a simple experiment to analyze whether our affinity computation captures fine-grained visual feature correspondences.
>
> ### *B. Experimental Verification*
> The overall procedure is similar to the visualization experiment shown in Figure 6. The experimental setup is as follows:
> 1.  First, we use a combination of a face segmentation model to extract a precise face mask for each character.
> 2.  Next, we use only the features from the `generated` character's **face region** as a query to attend to the entire feature map of the corresponding `reference` character.
> 3.  We then analyze the distribution of the resulting affinity scores. Specifically, we compare the **Area Ratio** (AR) (face mask area of the `reference` character / full character mask area of the `reference` character) to the **Affinity Ratio** (AF) (affinity score on the face region of the `reference` character / total affinity score for the `reference` character).
>
> | Metric | Mean | Variance |
> | :--- | :--- | :--- |
> | AF | 0.7874 | 0.0245 |
> | AR | 0.1038 | 0.0020 |
>
> On average, for the reference character, the facial region accounts for only **10.38%** of the area but holds **78.74%** of the total affinity. The `generated` character’s face shows a high affinity with the `reference` character’s face, while its affinity with the remaining area is much lower. This demonstrates that most of the affinity is concentrated in this fine-grained facial region, indicating that the affinity computation retains a significant level of fine detail.
>
> ## **II. Concern about Occlusion**
> ---
> Thank you for your critical question regarding occlusion handling. Our response consists of two parts: first, we will explain the reason of "abrupt changes", and second, we will show that our model still achieves a comprehensive SOTA performance on a public benchmark containing occlusion scenarios.
>
> ### *A. Cause of the Phenomena*
>
> Under severe occlusion, the accuracy of existing pose estimation algorithms can degrade dramatically, making it difficult to extract reliable driving signals. According to our automated data processing script, these unreliable skeletons are discarded. We thought these extreme cases would cause unpredictable behavior and interfere with the evaluation of the models' performance. The focus of our work is to evaluate model performance under reliable driving signals. To ensure fairness, this data processing pipeline was applied equally to all baselines. We recognize that handling such "abrupt changes" in driving signals caused by upstream tool is important for future research.
>
> ### *B. Performance on Occlusion Benchmark*
>
> Despite the aforementioned data processing challenges, our model SOTA performance on a **public** benchmark with frequent occlusions.
>
> We conducted a full quantitative evaluation on the **public multi-character benchmark from Follow-Your-Pose-V2 (FYP-V2)**, which contains frequent occlusions. As shown in **Appendix Table 1**, our method significantly outperforms all SOTA methods, including `Follow-Your-Pose-V2`.
>
> The complete quantitative comparison is as follows:
>
> | Method (Method) | SSIM ↑ | PSNR↑ | LPIPS ↓ | L1 ↓ | FID ↓ | FID-VID ↓ | FVD ↓ |
> | :--- | :---: | :---: | :---: | :---: | :---: | :---: | :---: |
> | DisCo | 0.793 | 29.65 | 0.239 | 7.64E-05 | 77.61 | 104.57 | 1367.47 |
> | DisCo+ | 0.799 | 29.66 | 0.234 | 7.33E-05 | 73.21 | 92.26 | 1303.08 |
> | MagicAnime| 0.819 | 29.01 | 0.183 | 6.28E-05 | 40.02 | 19.42 | 223.82 |
> | MagicPose| 0.806 | 31.81 | 0.217 | 4.41E-05 | 31.06 | 30.95 | 312.65 |
> | Animate Anyone| 0.795 | 31.44 | 0.213 | 5.02E-05 | 33.04 | 22.98 | 272.98 |
> | Animate Anyone*| 0.796 | 31.10 | 0.208 | 4.87E-05 | 35.59 | 22.74 | 236.48 |
> | Follow-Your-Pose-V2| 0.830 | 31.86 | 0.173 | 4.01E-05 | 26.95 | 14.56 | 142.76 |
> | **Everybody Dance (Ours)** | **0.879** | **32.49** | **0.151** | **0.92E-05** | **26.01** | **12.68** | **127.36** |
>
> ## **III. Concern about Related Dataset**
> ---
> Thank you for your valuable suggestion regarding these relevant datasets. We agree they are high-quality works and appreciate the opportunity to kindly explain what data is our task required:
>
> * **You2Me dataset:** You2Me contains high-quality and complex two-person interactions. However, the videos in this dataset are primarily filmed from a first-person perspective, meaning that typically only one complete person appears in the frame. The core of our work is to address identity confusion and preservation when multiple characters co-exist in the same frame. Therefore, the data paradigm of You2Me does not fully align with our "multi-character co-presence" task setting.
>
> * **InterHuman and DD100 datasets:** These datasets provide precise 3D human pose and shape parameters (e.g., SMPLx models). However, they primarily focus on the body's geometric structure and dynamics, lacking the detailed and diverse character appearances (e.g., clothing, hairstyles, faces) that are essential for our task. Our method aims to animate a character with a specific appearance from a reference image, which requires training data to contain both rich poses and their corresponding realistic appearance information. This was our original intention when we mentioned the "scarcity of high-quality multi-character video datasets" in our paper.
>
> Nevertheless, your suggestion is very helpful. We recognize that incorporating 3D priors into our pipeline is a valuable future direction, especially for challenges like severe occlusion. We will include this discussion about the datasets mentioned in the final version. Thank you again for the constructive feedback.
>
> ## **IV. Concern about Copyright and Privacy**
> ---
> Thank you for your important question regarding the privacy and copyright. We took this very seriously from the beginning. To construct and release our dataset and benchmark responsibly, we have strictly followed the principles below:
>
> ### *A. Privacy-First Manual Curation:*
>
> We did not engage in any automated or indiscriminate data scraping. Instead, all data was carefully and manually selected by our researchers according to strict criteria:
> *   **Source and Context:** We focused on videos created by professional performers (based on their subscriber count) on public platforms, filmed in public spaces like dance studios or stages. We purposely avoided content from private spaces, such as personal rooms. Besides, we didn't include any minors in our dataset.
> *   **Respect for Creators:** We strictly adhered to the creators' explicit wishes. Any video or creator account with statements like "Do not download," "No reposting," or "Not for AI training" was carefully excluded from our collection.
> *   **Data Minimization:** We followed the principle of data minimization at every stage. We only obtained information that was necessary to achieve our research objective. For instance, we extracted only the relevant video segments and discarded any unrelated content such as the creators’ profiles.
>
> ### *B. Protecting Creators with a 'Notice and Takedown' Policy*
>
> This policy is central to our ethical approach, aiming to protect and support anyone who appears in our video data.
> *   **No Raw Data Distribution:** We will never publicly release or redistribute original video files. For research purposes, we only provide links to publicly available sources.
> *   **Accessible Opt-Out Channel:** We provide a clear and straightforward way to contact us. Any creator appearing in a video can reach out at any time to request that we remove their data. We commit to immediately and permanently removing any related links from our dataset.
>
> ### *C. Restrictive Licensing for Academic Research:*
> We will release our curated dataset links, not the actual video data, under the Creative Commons CC BY-NC 4.0 license. This ensures the dataset is prohibited from being used for any commercial purposes by any individual or organization.
>
> We believe the measures we’ve outlined reflect our respect and effort to protect privacy and copyright, and that we do our best to advocate such respect and protection in the research community.

---

> ### Comment · Reviewer_Mm46 · 2025-08-04
>
> I appreciate the authors' efforts for strenghthening this work.
>
> After reading the authors' response, I realized that the occlusion issue is not focused in this paper, while it focuses on identity consistency. However, this also raises three new questions:
>
> - If no suitable labels can be found for occluded frames, do the authors have any potential methods to mitigate this issue?
>
> - If the method does not handle occlusion explicitly during training or testing, why does it outperform Follow-Your-Pose v2, which explicitly models depth to address occlusion, whereas the proposed method does not? Is this mainly due to the dataset? At the very least, the reasoning remains unclear. Also, which metric reflects performance under occlusion? Could EveryBodyDance consider incorporating depth modeling, similar to Follow-Your-Pose, to better handle occlusions? I feel it better to write the limitation clearly in revised version.
>
> - The authors mention that 3D data cannot be used for this task:
>
>     >“However, they primarily focus on the body's geometric structure and dynamics, lacking the detailed and diverse character appearances (e.g., clothing, hairstyles, faces) that are essential for our task.”
>
>     Although 3D models mainly use SMPL-X, their 3D motions can be applied to various character models. Is there a possibility for the authors to try rendering a dataset (even a small one) with different character models and backgrounds to generate a set of videos with ground-truth 2D labels? Wouldn't this help address both occlusion and diversity issues effectively?

---

> ### Author Response · Authors · 2025-08-04
>
> Thank you for your insightful questions and constructive feedback. We appreciate your deeper engagement with our work and would like to try to address your concerns.
>
> ## **I. Solving the Occlusion Challenge**
> ---
> You raise a critical point about occlusion handling. We agree this could limit our dataset's utility in some cases. Here are two promising directions we're considering:
> - **Depth-aware driving conditions**: Incorporating driving conditions with 3D information similar to DensePose could help solving occlusions.
> - **Segmentation-based approaches**: Leveraging segmentation masks (e.g., from SAM2) to design new driving conditions. Even when occlusion causes disconnected body parts (e.g., only head and legs visible), segmentation can still confirm they belong to the same character. However, removing shape-specific information from masks (e.g., distinctive hairstyles) remains a challenge.
> - **Point-tracking conditions**: Using point tracking methods (e.g., CoTracker 3) to specify where each character's body parts should appear during occlusions.
>
> ## **II. Performance Comparison with Follow-Your-Pose v2**
> ---
> Thank you for your suggestions. We will add this limitation concerning occlusion to the revised version. We will answer your questions from the following three aspects:
>
> ### *A. Reasons Behind Performance Gains*
> Follow-Your-Pose v2's excellent performance in their benchmark stems from two main aspects: (1) preventing feature contamination between characters, and (2) specifying which character's body parts should appear in front during occlusions. Regarding the first aspect, we also address feature contamination between characters. Regarding the second aspect, we conducted the following analysis:
>
> In occluded regions, our method randomly decides which character appears in front, but the result also remains clear and recognizable - it can still identify which reference character the body parts belong to, rather than producing distortion or deformation. This is primarily due to: (1) our method improves the decoupling capability for different reference characters' features, and (2) our dataset also contains cases of occlusion (during swap) where IDs can be correctly assigned. Besides, in FYP-V2's benchmark, these occluded regions usually constitute a **small** part of the characters' regions, so they may not significantly impact the overall video quality. Combined with our method's effective handling of feature contamination, this contributes to the better overall performance.
>
> Besides, Follow-Your-Pose v2 benchmark primarily comprises of scenarios where characters stand side-by-side with moderate occlusions. In these easier cases, our method can still extract accurate skeleton poses and assign correct ID labels. However, the severely occluded cases we discussed in the original question are not involved in Follow-Your-Pose v2's benchmark.
>
> ### *B. Metrics Reflecting Performance under Occlusion*
> All listed metrics (used in FYP-v2 benchmark) reflect overall image/video quality.  In future work, developing specialized metrics that reflect the fidelity of occluded regions is also an important research topic.
>
> ### *C. Potential for Incorporating Depth Modeling*
> Your suggestion to incorporate depth modeling similar to Follow-Your-Pose v2 is constructive, and our method would certainly benefit from this enhancement. The interesting and crucial challenge is that severely occluded frames often yield deteriorated or missing skeletons, while FYP-v2 relies on skeletons to create Depth Order maps through dilation. Developing robust depth modeling methods that can handle such challenging cases represents a valuable research direction in the future.
>
> ## **III. Utilizing 3D Data for Dataset Generation**
> ---
> Thank you for your inspiring suggestion. Indeed, it can potentially benefit our model by leveraging 3D model rendering for diverse appearance and motions with accurate gt-labels to enhance our datasets.  This could effectively help to address both occlusion and diversity challenges. We will include this valuable perspective in the future research part in our revised version.
>
> Thank you again for these thoughtful suggestions, which will certainly inspire our future research directions. We look forward to further discussions with you.

---

> ### Author Response · Authors · 2025-08-08
>
> Dear Reviewer,
>
> Thank you for your constructive guidance, especially your insightful suggestion about leveraging 3D datasets to handle severe occlusion. These perspectives open new possibilities for our future research.
>
> We sincerely cherish this opportunity to learn from your expertise. If you have any further insights or remaining concerns, we would be happy to hear them and provide clarification.
>
> Thank you again for your helpful feedback.
>
> Best regards,
>
> The Authors

---

> > ### Comment · Reviewer_Mm46 · 2025-08-09
> >
> > Thanks for the discussion. The author has addressed my questions.

---

> > > ### Author Response · Authors · 2025-08-09
> > >
> > > Dear Reviewer,
> > >
> > > We are delighted that we could address your questions. This discussion will greatly benefit our future research! Thank you for your time and support!
> > >
> > > Best regards,
> > >
> > > The Authors

---

### Official Review · Reviewer_QHRK · 2025-07-02

**Clarity:** 2
**Significance:** 3
**Originality:** 2
**Rating:** 4
**Confidence:** 4

**Summary:**

The paper introduces a multi-identity feature association graph during the training phase to address the issue of confusion in multi-person video generation, and builds corresponding evalution benchmark. In detail, a mask query attention is proposed to optimize the identify graph. The experiments demonstrate the outperformances in identify matching and visua fidelity

**Questions:**

1. Lack of related work listed above.

2. Instead of using ICG, a more straightforward approach, similar to the method described in "Blending Custom Photos with Video Diffusion Transformers," could be to use the softmax results of the attention between the reference identity features and the target features?

3.The foundational video generation models have evolved significantly in recent times. It would be valuable to explore this topic based on models like Wan2.1 or HunyuanVideo, or even a purely text-to-image model

**Ethical Concerns:**

["NO or VERY MINOR ethics concerns only"]

**Final Justification:**

The authors have provided the required reference papers and comprehensive comparative experimental results. Furthermore, they have supplemented their work with experiments under single-person dancing scenarios, which further validate the effectiveness of their algorithm. The experimental results convincingly demonstrate the superiority of the proposed method. I also acknowledge that multi-person dance generation is an interesting direction for future research. Overall, my final recommendation is a borderline accept.

**Limitations:**

It is recommended to supplement experimental comparisons on single-person videos to demonstrate the model's generalization performance, and what about more than two persons?

**Quality:**

2

**Strengths And Weaknesses:**

Strength
This paper is a systematic solution for ensuring Identity Correspondence correctness in multi-character animation, a challenging problem when extending pose-driven animation to multi-character settings with position swaps. and the benchmark is provided to facilitate academic progress

Weakness
The highly relevant papers listed below are not mentioned, and these methods should be analyzed or compared accordingly. The performance analysis of single-id video is also necessary in the paper.
a、Follow-Your-Pose v2: Multiple-Condition Guided Character Image Animation for Stable Pose Control
b、DanceTogether! Identity-Preserving Multi-Person Interactive Video Generation
c、Identity-preserving text-to-video generation by frequency decomposition
d、Blending Custom Photos with Video Diffusion Transformers

---

> ### Author Rebuttal · Authors · 2025-07-27
>
> ## **I. Concern about Missing Related Work**
> ---
> Thank you for your valuable feedback on missing related works. We divide the four related work you mentioned into two categories: (1) works on **Pose-Driven Character Animation**, which address the same task as ours, and (2) works on **ID-Preserving Text-to-Video (T2V)**, which tackle a different problem. We will cite and discuss these four excellent works in our final version.
>
> ### *A. Works on Pose-Driven Character Animation*
>
> Pose-driven character animation aims to generate a video by driving a source character with a reference image and a pose sequence. However, the core contribution of our method is **spatial controllability** — determining *which character appears at which location*.
>
>
> #### *a. "Follow-Your-Pose v2" (FYP-v2)*
>
> We appreciate FYP-v2's valuable exploration in tackling multi-character animation task. We kindly invite you to check the reference and comparison of FYP-v2 in our Appendix E.1.
>
> For a fair comparison, we also evaluate our method on the **official** FYP-v2 benchmark, which focuses on occlusion scenarios without involving position swapping. As shown in **Appendix E.1, Table 1** of our supplementary material, our method outperforms all baselines across all metrics. This indicates that our approach not only offers spatial controllability but also achieves state-of-the-art performance in occlusion scenarios. We also listed comparison results with FYP-v2 below.
> | Method | SSIM↑ | PSNR↑ | LPIPS↓ | FID↓ | FID-VID↓ | FVD↓ |
> | :--- | :---: | :---: | :---: | :---: | :---: | :---: |
> | FYP-V2 | 0.830 | 31.86 | 0.173 | 26.95 | 14.56 | 142.76 |
> | **Ours** | **0.879** | **32.49** | **0.151** | **26.01** | **12.68** | **127.36** |
>
> Furthermore, it is candidly acknowledged that  handling "position swap" remains a limitation of their method in **Appendix D** of FYP-v2 paper. Our work addresses this problem. As shown in the Figure 7, by swapping the ID in the input skeleton, our model correctly generates the characters in their new, swapped positions while maintaining their unique appearances.
>
> #### *b. "DanceTogether" (DanceTog)*
>
> As a concurrent work, DanceTog was released on arXiv after our submission deadline and without publicly available code, so it was not included in our original discussion. We believe that our method and DanceTog represent two distinct yet equally valuable technical approaches. The core of DanceTog lies in its `MaskPoseAdapter`, which integrates mask and pose information at the input level to provide strong guidance. We make contributions at both the input and supervision levels: Identity-Embedded Guidance(IEG) is used to provide ID instruction and pose sequence, while Identity Matching Graph (IMG) supervises the model to generate according to the ID instructions from IEG.
>
> Our core advancement lies in **scalability and flexibility**. As noted in **Appendix A** of the DanceTog paper, extending their method to more than two characters leads to significant computational overhead. In contrast, our approach incurs no additional inference cost when handling more than two characters (see **Figure 4** in our Appendix E.5), highlighting our method's scalability. Furthermore, as shown in **Figure 7** of our main paper, our method also exhibit strong flexibility (swapping character positions by changing the skeleton).
>
> ### *B. Regarding Works on ID-Preserving T2V*
>
> ID-Preserving T2V models create scenes from text while embedding an ID, without precise control over position and pose. In contrast, pose-driven character animation requires maintaining visual elements from the entire reference image, with precise control over character position and pose. Despite the differences in task formulation, we recognize that discussing the underlying methodologies remains valuable.
>
> * `ConsisID`'s core idea about frequency decomposition to solve **single-ID preservation** problem is inspiring. However, it doesn't involve identity correspondence between multiple characters.
> * `Ingredients` (Blending Custom Photos with Video Diffusion Transformers): It handles multi-ID scenarios using a pixel-level classification loss through an ID Router, delivering impressive performance on multi-ID video customization. In contrast, our method constrains character-level identity correspondence by capturing model's native attention maps to enforce generating the correct character at the correct position. This is a very interesting and insightful comparison, and we will include more detailed discussion in the final version.
>
> ## **II. Suggestion on Affinity Computation**
> ---
> Thank you for your inspiring suggestion. Given the differences between the two tasks, directly transferring the `Ingredients` method is difficult. `Ingredients` incorporates per-character CLIP embeddings with a text prompt for **relative** position control. In contrast, the pose-driven character animation task requires generating the correct character at the correct position and is guided by pose for precise motion and **absolute** position control. Before explaining why we adopted the IMG paradigm, we designed two variants of our method based on `Ingredients`. If there are any detailed setting you'd like us to implement, please further discuss with us. Here is our implementation details:
>
> * `CrossAttn-Based`: We segment the reference image to extract `m` distinct identity features using a CLIP encoder for each character. In the Cross Attention layer of the model (**Figure 3**), instead of interacting with the CLIP embedding of the whole image as usual, we compute the affinity between the features of the generated frame and the `m` CLIP embeddings for each character. After applying softmax, we obtain an `[HW, m]` affinity map, where `H` and `W` are the latent shape. Each entry indicates the probability of a latent patch belonging to one of the `m` reference characters, with the background masked out.
>
> * `Sim-Based`: This approach directly measures similarity between characters. We use masks to extract features for `m` reference characters and `n` generated characters from the model's intermediate feature. By computing their dot-products and applying softmax, we generate an `[n, m]` affinity matrix, where each entry `(i, j)` indicates the similarity between the `i`-th generated character and the `j`-th reference character.
>
> Each variant is optimized using cross-entropy loss for classification.
>
> |Method | SSIM ↑ | PSNR* ↑ | LPIPS ↓ | FID ↓ | FID-VID ↓ | FVD↓ |
> |:---|:---|:---|:---|:---|:---|:---|
> | CrossAttn-Based | 0.648 | 15.69 | 0.315 | 45.82 | 29.22 | 306.75 |
> | Sim-Based | 0.629 | 16.25 | 0.329 | 47.09 |24.35| 269.14 |
> | **Ours** | **0.654** | **16.93** | **0.304** | **40.19** | **23.58** | **225.06** |
>
> We chose IMG+MQA for affinity computation as it directly utilizes the model's native attention scores, tightly coupled with the generation process. In contrast, `CrossAttn-Based` relies on additional features unrelated to the model’s behavior, while `Sim-Based` struggles to distinguish visually similar characters—issues our method inherently avoids. However, this is a very inspiring suggestion, and we are willing conduct further experiment in this direction.
>
> ## **III. Concern about Different Number of Characters**
> ---
> Thank you for your critical questions regarding our model's generalization capabilities. Our method demonstrates superior generalization on both single-character and multi-character (N>2) video generation, outperforming strong baselines in both scenarios. Due to space constraints, we kindly invite you to check some details in our response to Reviewer `acUv`.
>
> ### *A. Generalization to Single-Character Task*
>
> Our method not only maintains performance on single-character tasks but also improves upon the single-character backbone (AnimateAnyone's public version implemented by Moore Threads), demonstrating its generalization capability. As detailed in our response to Reviewer `acUv`, we conducted a comprehensive ablation study. Below are the results on the **public** single-character TikTok (Jafarian et al., 2021) benchmark.
>
> | Method | PSNR*↑ | LPIPS↓ | SSIM↑ | FID↓ | FVD↓ | FID-VID↓ |
> | :--- | :--- | :--- | :--- | :--- | :--- | :--- |
> | AnimateAnyone | **17.85** | 0.280 | 0.768 | 52.15 | 209.14 | 25.864 |
> | **Ours** | 17.78 | **0.279** | **0.772** | **40.56** | **163.85** | **20.294** |
>
> ### *B. Generalization to More Characters*
>
> Our method effectively scales to multi-character scenes, as shown by the qualitative results in **Appendix Figure 4** of the supplementary materials. We also present quantitative results for 3-5 character scenarios (see Reviewer `acUv` for details), demonstrating that our method consistently outperforms baselines in these challenging cases.
>
> | Method | PSNR*↑ | LPIPS↓ | SSIM↑ | FID↓ | FVD↓ | FID-VID↓ |
> | :--- | :--- | :--- | :--- | :--- | :--- | :--- |
> | AnimateAnyone | 14.70 | 0.370 | 0.606 | 56.66 | 401.51 | 35.356 |
> | AnimateAnyone*| 14.92 | 0.363 | 0.607 | 64.34 | 406.08 | 36.308 |
> | UniAnimate | 15.62 | 0.338 | 0.640 | 57.75 | 348.91 | 29.030 |
> | **Ours** | **16.68** | **0.315** | **0.671** | **42.84** | **261.01** | **23.571** |
>
> We will integrate these results into the final version.
>
> ## **IV. Suggestion about Foundational Models**
> ---
> Thank you for your insightful suggestion. Leveraging foundational models is indeed a promising area for future research. Our chosen backbone, AnimateAnyone, is built on the text-to-image model Stable Diffusion 1.5, aligning with your perspective on using such models. Extending our method to the latest video generation models is a valuable direction for future exploration. In the future, we are open to transferring our method to more advanced foundational models.

---

> > ### Comment · Reviewer_QHRK · 2025-08-08
> >
> > The authors have provided the required reference papers and comprehensive comparative experimental results. Furthermore, they have supplemented their work with experiments under single-person dancing scenarios, which further validate the effectiveness of their algorithm. The experimental results convincingly demonstrate the superiority of the proposed method. I also acknowledge that multi-person dance generation is an interesting direction for future research. Overall, my final recommendation is a borderline accept.

---

> ### Author Response · Authors · 2025-08-07
>
> Dear Reviewer,
>
> Thank you for your thoughtful review. Your suggestions, such as different ways for affinity computation, were particularly inspiring, which led us to conduct some interesting experiments.
>
> We hope our responses and additional experiments have addressed your concerns. We understand this is a busy time with multiple reviews to consider. If you have a moment in the remaining two days of discussion, we would be grateful for any feedback on whether our clarifications were helpful.
>
> Best regards,
>
> The Authors

---

> ### Author Response · Authors · 2025-08-08
>
> Dear Reviewer,
>
> Thank you so much for taking the time to review our responses. We truly appreciate your recognition of our additional experiments and comparisons.
>
> We are delighted that our clarifications addressed your concerns. Your suggestions have significantly strengthened our paper, and we're extremely grateful for your recognition of our work!
>
> Thank you again for your engagement with our work!
>
> Best regards,
>
> The Authors

---

### Official Review · Reviewer_SZsZ · 2025-07-02

**Clarity:** 3
**Significance:** 2
**Originality:** 3
**Rating:** 4
**Confidence:** 4

**Summary:**

The paper proposes a method to address the identity preserving issue in multi-character animation video generation. It proposes a bipartite matching between reference character and generated character frame-wisely. By formulating the correspondence matching into a matching loss and applied during training, the method is able to better maintain identity in the generation and outperform existing approaches consistency under different settings. Experiments show improved generated video quality on the proposed ICE benchmark.

**Questions:**

1.  **Instance segmentation**

In order to segment the generated character in each generated frame, does that mean during each training iteration, the method has to run segmentation on the generated results? In the early state training, the generation could be poor and how to ensure it could segment the generated character correctly. Also, what is training overhead by applying such segmentation per iteration.

2. **How to determine the number of person**

I would like to know how to determine m and n, the number of characters for reference and generated. For different inputs with different number of person, how does the current method count number of character in the generation process and do the matching.

3. **Occlusion**

I am curious how does the IMG address occlusion issue in occluded frames, without consider temporal information, the per-frame setting will suffer from bad matching for occluded frames.

**Ethical Concerns:**

["NO or VERY MINOR ethics concerns only"]

**Final Justification:**

After reading the rebuttal, I believe the authors have addressed most of my concerns. The experimental results on both multi-character and occlusion datasets outperform all other methods across all metrics. However, I still have some reservations about the evaluation, as it relies on indirect metrics that are not fully convincing (not directly judge the character consistency).  The method achieves the best results even without explicitly handling occlusion, which raises slight questions about the robustness of the evaluation.

The original paper lacks experiments, while the rebuttal gives sufficiently large experiments and all those tables should be reported in the final version. Considering this, I will raise my score to borderline accept.

**Limitations:**

yes

**Quality:**

2

**Strengths And Weaknesses:**

Strengths

**Important task**

Maintain identity consist across videos is a important task. The paper does the weakness analysis of existing methods and show the importance of addressing such issue in multi character animation.


**Good quantitative results compare to existing approaches**

The method outperform other methods under the same setting. I appreciate the author conducts a strict comparison and remove the influence of different datasets.

Weakness

**No temporal correspondence in the proposed IMG**

The bipartite graph is constructed per frame and there is no consistent person match across time, which loss the temporal consistency. I would like to know why not consider the temporal information in IMG and how current IMG could improve the temporal consistency by only considering the matching framewisely.

**Affinity computation**

I am not fully convinced the affinity could reflect the person identity information. As the feature contains the pose information, how to ensure the affinity score has remove the pose information for both reference and generated character.

**Scaling person**

I understand the current dataset doesn't contain too much person. But i am highly doubt the proposed method could work for larger number of person input. As most of results only showcase two person (which is quite simple in matching), I wonder how does the paper work for a group of person and does the training still applicable to such larger number, as the current training needs instance segmentation per iteration.

---

> ### Author Rebuttal · Authors · 2025-07-25
>
> ## **I. General Response**
> ---
> Thank you for your valuable feedback. You've raised excellent points that helps us clarify our system design. Our method is a **systematic solution**. Our core innovation is the Identity Matching Graph (IMG). However, it needs to cooperate with other components (e.g., Identity-Embedded Guidance), which are also designed to address some of important challenges you concerned. The core of our task, identity correspondence for multi-character animation, lies in **spatial controllability**—*determining which character appears at which location and performs what pose*. This differs from identity preservation.
>
> ## **II. Concern about Temporal Correspondence**
> ---
> Thank you for your valuable question. To clarify upfront, this key to achieve temporal consistency is not handled by IMG, but a simple yet critical component of our system: the **Identity-Embedded Guidance (IEG)**. Your observation is correct: our frame-wise Identity Matching Graph (IMG) doesn't handle this role directly. Its purpose is instead to supervise the model's ability to follow the IEG's identity guidance during training:
> * **What IEG is:** IEG is an identity guidance signal that embeds a **persistent** Identity(ID) (represented by different color) into the pose skeleton. This ID remains consistent for a specific character throughout the whole video, even if the characters swap positions. As shown in our **Appendix's Figure 1**.
> * **When IEG is Used:** IEG is utilized as the guidance signal during both training and inference.
> * **What IEG Offers:** A reference IEG specifies each character's ID in the reference image, and a target IEG sequence provides the corresponding motion trajectory for each character.
> * **How IEG works:** IEG guides the model to *generate the correct character at the correct position*. As shown in the **Figure 7**, by swapping the ID in the input target IEG sequence, our model correctly generates the characters in their new, swapped positions while maintaining their unique appearances.
> * **How IEG collaborates with IMG:** During training, the IEG functions as the **ID instructor**, providing identity labels and pose guidance, while the IMG serves as a **supervisor**, ensuring the model learns to follow the IEG's ID instruction.
> * **How IEG generated**: IEG is generated using cross-frame identity tracking and pose estimation (see our Appendix B.1).
>
> | Method            | SSIM↑     | LPIPS↓     | FID↓     | FVD      |
> | ----------------- | --------- | ---------- | -------- | -------- |
> | w/o Reference IEG | 0.619     | 0.331      | 42.84    | 251.05   |
> | w Reference IEG   | **0.654** | **0.304**  | **40.19** | **225.06** |
>
> We designed a simple experiment. When using the same model for inference, one version does not mark the identity of the reference character (`w/o Reference IEG`), while another version does (`w Reference IEG`). We see that without identity instruction, the model’s performance decreases significantly, especially on the FVD metric, which reflects temporal coherence.
>
> Furthermore, the **Temporal Attention** blocks within our model provide temporal smoothness during the generation process. However, we agree that your suggestion about incorporating temporal correspondence within the IMG is a promising direction for future research.
> ## **III. Concern about Instance Segmentation**
> ---
> Thank you for your reasonable question regarding our implementation about segmentation.
>
> To clarify directly, the segmentation is **not** performed on the `generated` image during training or inference. Instead, it is performed offline on the ground-truth (GT) `target` and  `reference` image from the training video samples. During training, the `target` image serves as the GT of the  `generated` image. Therefore, we only use the clean, pre-computed mask of the `target` image. Since this is a one-time, offline pre-process procedure of the dataset, the training loop is free from expensive SAM2 segmentation **computational overhead**. Furthermore, the segmentation **accuracy** is reliable because we only extract masks from GT `target` frames, not the potentially poor `generated` images.
>
> ## **IV. Question about Determining the Number of Persons**
> ---
> Thank you for this critical question. We will clarify how the number of characters is determined during the construction of the Identity Matching Graph (IMG). For IMG construction, it **dynamically** determines node number by the training sample:
>
> - `m` is the number of skeletons for the reference image, each paired with a pre-computed unique SAM2 mask for IMG node construction.
> - `n` is the number of skeletons for the target image, each also paired with a pre-computed unique SAM2 mask for IMG node construction.
>
> In short, `m` and `n` are determined by the number of skeletons, without an explicitly counting mechanism.  A case can be found in **Appendix's Figure 4**, where the target IEG contains three skeletons, our model adaptively generates three characters. As for "**how to do the matching**", IMG is dynamically constructed from various `m` and `n`. If you have any further questions, we are willing to discuss then with you.
>
> ## **V. Concern about Scaling Person Number**
> ---
> Thank you for this crucial question on scalability. You are correct that our training data consists mainly of two-person scenes, which makes generalization to more characters critical test for our method.
>
> ### *A. Generalization to More Characters*
> Our Appendix includes a qualitative result on a challenging three-person scene (**Appendix Figure 4**). To provide comprehensive quantitative proof, we further conducted new experiments on a challenging benchmark comprising **3-5 characters** scene. The results below confirm that our method's effectiveness and generalization performance:
>
> | Method | PSNR*↑ | LPIPS↓ | SSIM↑ | FID↓ | FVD↓ | FID-VID↓ |
> | :--- | :--- | :--- | :--- | :--- | :--- | :--- |
> | AnimateAnyone | 14.70 | 0.370 | 0.606 | 56.66 | 401.51 | 35.356 |
> | AnimateAnyone*| 14.92 | 0.363 | 0.607 | 64.34 | 406.08 | 36.308 |
> | UniAnimate | 15.62 | 0.338 | 0.640 | 57.75 | 348.91 | 29.030 |
> | **Ours** | **16.68** | **0.315** | **0.671** | **42.84** | **261.01** | **23.571** |
>
> Furthermore, a key advantage of our framework is that this scalability comes with no additional computational overhead at inference time, as the Identity Matching Graph (IMG) is only constructed during training.
>
> ### *B. Does the Training Still Applicable?*
> Yes, the training process remains fully applicable. As discussed in our response to Question IV, the construction of the IMG is an entirely dynamic process. The number of nodes for any given training sample is determined by the number of character masks pre-computed from that video clip. Consequently, our training framework supports a variable number of characters without modifications.
>
> ## **VI. Concern about Affinity Disentanglement**
> ---
> Thank you for this question, as it touches upon the core of our learning mechanism. First, let's assume that the "pose" you mentioned is the "posture" such as sit or stand. If this is not the case, please don't hesitate to further inform us. We don't explicitly remove pose information from the features. Instead, our training framework is designed to **mitigate** its influence.
>
> This is demonstrated in **Figure 6** (third row). In this example, both reference characters have similar postures. If posture information were the important factor, the affinity map would show a strong response for both. However, as can be seen, our method's attention is sharply focused on the correct reference character, proving that our training effectively weakens the influence of pose.
>
> If the model were to rely on posture similarity and assign high affinity to a wrong reference character, our IMG loss (`L_match`) would penalize this. Therefore, to minimize the loss, the model is forced to follow the IEG's identity instruction (as we discussed in Response II), downplaying the role of posture information.
>
> ## **VII. Concern about Occlusion**
> ---
> ### *A. How to Operate*
>
> Thank you for your valuable question on occlusion. During training, IMG rely on ground-truth masks, which can be extracted even when characters are occluded. Our IEG provides the correct identity label for that character, and the IMG uses this information to compute a matching loss. IMG teaches the model to associate an occluded appearance with the correct, full identity from the reference image. We agree that your suggestion to incorporate more direct temporal modeling is an excellent direction for future work, especially for handling severe occlusion.
>
> ### *B. Performance on Occlusion Benchmark*
>
> We quantitatively validated our method's performance on the **public** Follow-Your-Pose-V2 benchmark, which is specifically noted for its frequent and challenging inter-person occlusions. As reported in the **Appendix (Table 1)**, EverybodyDance achieves SOTA results, outperforming all baseline methods across quality metrics on this benchmark.
> | Method | SSIM↑ | PSNR↑ | LPIPS↓ | L1↓ | FID↓ | FID-VID↓ | FVD↓ |
> | :--- | :---: | :---: | :---: | :---: | :---: | :---: | :---: |
> | DisCo| 0.793 | 29.65 | 0.239 | 7.64E-05 | 77.61 | 104.57 | 1367.47 |
> | DisCo+| 0.799 | 29.66 | 0.234 | 7.33E-05 | 73.21 | 92.26 | 1303.08 |
> | MagicAnime| 0.819 | 29.01 | 0.183 | 6.28E-05 | 40.02 | 19.42 | 223.82 |
> | MagicPose| 0.806 | 31.81 | 0.217 | 4.41E-05 | 31.06 | 30.95 | 312.65 |
> | AnimateAnyone| 0.795 | 31.44 | 0.213 | 5.02E-05 | 33.04 | 22.98 | 272.98 |
> | AnimateAnyone†| 0.796 | 31.10 | 0.208 | 4.87E-05 | 35.59 | 22.74 | 236.48 |
> | FYP-V2 | 0.830 | 31.86 | 0.173 | 4.01E-05 | 26.95 | 14.56 | 142.76 |
> | **Ours** | **0.879** | **32.49** | **0.151** | **0.92E-05** | **26.01** | **12.68** | **127.36** |

---

> > ### Comment · Reviewer_SZsZ · 2025-08-05
> >
> > Thanks the author for clarification and addressing my questions. Here are some concerns.
> >
> >
> > 1. IEG
> >
> > It is not surprising that a model trained with IEG will behave poorly at inference by removing it. My concern is IEG seems contain both spatial and temporal correspondences and cover all necessary information of IMG, why in table 1, the well-trained end-to-end model behaves poorly, I would like to see some comparison or failure modes of this baseline (supp videos only compare against different methods). Another question is given we have the complete information, is it equivalent to do some contrastive learning on the masked region to increase discrepancy as the IMG did, what benefit does IMG provides compare to IEG?
> >
> >
> > 2. Image quality
> >
> > I am curious why IMG could also improve individual image quality? As IMG apply a stricter constraint on weights during training, will the final model have large improvement on FID than other baseline or models without applying such harder constraint.
> >
> > 3. More Characters
> >
> > I wonder are the results zero-shot performance for each method?

---

> ### Author Response · Authors · 2025-08-05
>
> Thank you for these valuable questions. We appreciate your deeper discussion with our work and would like to address your concerns.
>
> ## **I. Questions regarding IEG**
> ---
> You correctly point out that IEG contains both spatial and temporal correspondences to establish identity relationships. Our initial approach was exactly to combine IEG with end-to-end training to help the model learn correct identity correspondences.
>
> ### *A. Failure Modes of End-to-End Models*
> As you observed in the Table 2, the end-to-end model performs below expectations (though still better than vanilla finetuning AnymateAnyone without IEG). **Figure 5** shows some failure modes—when character positions are swapped, the end-to-end model fails to handle identity correspondence. This was also the challenge we initially struggled with. Despite IEG provides complete identity information, we found in practice that: (1) it is challenging for model to recognize color encoding as an identity instruction through end-to-end training; (2) global L2 loss provides insufficient supervision for local character regions. Even with additional constraints on the masked region (the End2End-M variant), the problem remains unsolved.
>
> ### *B. Core Advantages of IMG over End-to-End Training*
> IMG provides a stronger supervisory mechanism compared to end-to-end training to ensure the model follows IEG's identity instructions. The key difference is that IMG explicitly constrains the model's **native attention patterns**, not just the final output. By explicitly modeling identity correspondence, IMG ensures the model focuses on the correct reference character (marked by IEG) when generating each character. The attention visualizations in **Figure 6** demonstrate this phenomenon. With IMG, the model's attention focuses on the correct IEG-marked characters, while the end-to-end approach produces scattered or incorrect attention patterns.
>
> ### *C. Connection and Comparison to Contrastive Learning*
> Your observation about contrastive learning on the masked region is very perceptive. IMG implements a similar mechanism—increasing discrepancy between correct and incorrect identity correspondences through explicit modeling.
>
> Contrastive learning typically requires increasing feature differences between different classes. In our scenario, when two characters have similar appearances (e.g., people wearing similar clothing), contrastive learning would force their features apart, potentially compromising their natural visual similarity. IMG enforces correct identity correspondence by supervising attention weights. Essentially, IMG focuses on "which character to reference during generation" rather than "making characters look different." This constraint might be better suited for handling characters with **similar** appearances but **different** identities, allowing accurate identity correspondence while preserving their original appearance.
>
> This is a very interesting question and we'd be happy to explore it further with you.
>
> ## **II. Questions regarding Image Quality**
> ---
> Thank you for raising this interesting point. IMG's improvement in image quality comes from two main aspects.
> First, by achieving effective identity correspondence, IMG ensures characters are generated in their correct positions, improving overall visual quality metrics (e.g., FID). In contrast, other methods without such strict constraints often struggle with identity correspondence and generate characters in wrong positions, especially when characters have notably different appearances, it can significantly affect quantitative metrics. Second, IMG focuses primarily on character regions without compromising background stability. This targeted approach allows IMG to improve image quality without sacrificing the overall scene coherence. The combination of these factors results in better FID scores, as demonstrated in Table 1 and Table 2.
>
> ## **III. Question regarding More Characters**
> ---
> Yes, the results for three or more characters are zero-shot performance, as our training dataset only contains videos with two characters.
>
> We hope we've addressed your concerns and welcome any further discussion.

---

> ### Author Response · Authors · 2025-08-08
>
> Dear Reviewer,
>
> Thank you for the enlightening discussion. Your feedback has encouraged us to view our work from new perspectives.
>
> We deeply cherish this opportunity to communicate with you. If any of our responses need further clarification or if new concerns arise, it would be our pleasure to explore them with you.
>
> Thank you again for your time.
>
> Best regards,
>
> The Authors

---

### Official Review · Reviewer_acUv · 2025-07-02

**Clarity:** 3
**Significance:** 3
**Originality:** 3
**Rating:** 4
**Confidence:** 2

**Summary:**

This paper aims to extend the one-character pose-driven animation task to multi-characters and propose the EverybodyDance method to solve this problem.  EverybodyDance explicitly builds the character correlation between the reference image and target video clips instead of simply implicitly mapping like ReferenceNet. Specifically, EverybodyDance introduces the Identity Matching Graph (IMG) to connect the characters from different sources and uses character latent features to represent the node and Mask–Query Attention (MQA) to represent the match weights of each character pair.  Other techniques like identity-embeded guidance, a multi-scale matching method and pre-classified sampling are proposed to improve model robustness and IC accuracy. Finally, a new Multi-Character Animation  benchmark dataset is introduced to evaluate the effectiveness of EverybodyDance  and other baselines. EverybodyDance achieves superior performance than other baselines.

**Questions:**

Please mainly address the questions in the weakness section.

**Ethical Concerns:**

["NO or VERY MINOR ethics concerns only"]

**Final Justification:**

The rebuttal well addressed some of my concerns. After a few rounds of discussion with the authors, most of concerns are resolved. Thus, I would raise the final rating to borderline accept.

**Limitations:**

The paper slightly mentioned the limitations in the end of the paper.

**Quality:**

2

**Strengths And Weaknesses:**

# Strengths:
  - This paper proposes an explicit identity mapping method to solve the task of multi-character animations.
  The key techniques Identity Matching Graph (IMG) and Mask-Quey Attention(MQA) sound novel and reasonable.
  - The quantity and quality results in a multi-character benchmark demonstrate the effectiveness of EverybodyDance.

# Weaknesses:
  - To my best of knowledge, EverybodyDance is not the first paper to solve the Multi-Character pose-driven Animation task. Previous work like [1] has proposed a method to solve this task. More discussions and comparisons are necessary to added to the main paper.
  - From the visualization of figures in this paper, there are almost two characters in reference images and target video clips. Does the training and test dataset consist of samples with only two characters? Can EverybodyDance also be extended to reference images of more than 2 characters? More distribution descriptions about the multi-character dataset and target visualization would be better to show the effectiveness of EverybodyDance.
  - EverybodyDance is finetuned with a custom multi-character dataset based on the AnimateAnyone and achives better performance than baselines in ICE benchmark. Because of the finetuning paradigm, we doubt that the FT process may introduce multi-character  bias and further destroy generalization for reference images with single characters. It is also necessary to make comparisons in test dataset involving single characters.


[1] Xue J, Wang H, Tian Q, et al. Follow-Your-Pose v2: Multiple-Condition Guided Character Image Animation for Stable Pose Control[J]. arXiv preprint arXiv:2406.03035, 2024.

---

> ### Author Rebuttal · Authors · 2025-07-25
>
> ## **I. Concern about Missing Related Work**
> ---
> Thank you for pointing out this important related work and for emphasizing the necessity of a direct comparison. We completely agree that this discussion was not highlighted sufficiently in the main body of our initial submission. We are grateful for your feedback, as it gives us the opportunity to clarify the detailed comparison and core differences between our work and "Follow-Your-Pose v2" (FYP-v2).
>
> ### *A. On Experimental Comparison:*
>
> We directly compared with FYP-v2, as shown in **Appendix E.1 (Table 1 and Figure 2)** of supplementary material. Although the official code and weights for Follow-Your-Pose v2 have not been released, their publicly available benchmark enabled us to perform a fair and direct comparison. For your convenience and clarity, we summarized the comparison results below. As shown in the table, our method achieves superior performance on FYP-v2's official multi-character benchmark across all image and video quality metrics.
>
> | Method | SSIM↑ | PSNR↑ | LPIPS↓ | L1↓ | FID↓ | FID-VID↓ | FVD↓ |
> | :--- | :---: | :---: | :---: | :---: | :---: | :---: | :---: |
> | DisCo| 0.793 | 29.65 | 0.239 | 7.64E-05 | 77.61 | 104.57 | 1367.47 |
> | DisCo+| 0.799 | 29.66 | 0.234 | 7.33E-05 | 73.21 | 92.26 | 1303.08 |
> | MagicAnime| 0.819 | 29.01 | 0.183 | 6.28E-05 | 40.02 | 19.42 | 223.82 |
> | MagicPose| 0.806 | 31.81 | 0.217 | 4.41E-05 | 31.06 | 30.95 | 312.65 |
> | AnimateAnyone| 0.795 | 31.44 | 0.213 | 5.02E-05 | 33.04 | 22.98 | 272.98 |
> | AnimateAnyone†| 0.796 | 31.10 | 0.208 | 4.87E-05 | 35.59 | 22.74 | 236.48 |
> | FYP-V2 | 0.830 | 31.86 | 0.173 | 4.01E-05 | 26.95 | 14.56 | 142.76 |
> | **Ours** | **0.879** | **32.49** | **0.151** | **0.92E-05** | **26.01** | **12.68** | **127.36** |
>
> ### *B. On the Fundamental Difference in Problem Formulation:*
>
> Beyond the advantages in quantitative metrics, we wish to further highlight the **fundamental difference** in problem formulation between our work and FYP-v2, which constitutes our core innovation. The challenges of multi-character animation can be divided into two levels:
>
> * Level 1: Feature Interference: This refers to avoiding interference of appearance features (e.g., clothing) when multiple characters are positioned side-by-side in the same frame. FYP-v2 made significant progress in addressing this challenge through its depth order guider.
>
> * Level 2: **Spatial Controllability:** This refers to a model's fundamental ability to control *which* character appears at *which* spatial location. A failure in this controllability could lead to the most critical error: e.g., character A appearing with character B's appearance after they swap positions. This is a more profound and difficult challenge than mere feature interference.
>
> Our work, "Everybody Dance," is the **first** framework to systematically address this Level 2 challenge of spatial controllability. Our primary contribution is ensuring robust Identity Correspondence (IC), accurately tracking and controlling character identities as their spatial layout changes.
>
> In fact, it is candidly acknowledged that spatial controllability remains a limitation of their method in **Appendix D of FYP-v2 paper**. Our work directly addresses this key problem that the prior work didn't cover.
>
> In summary, our work not only surpasses FYP-v2 in performance on its own benchmark, but, more importantly, defines and systematically solves the profound challenge of spatial controllability. We will integrate this important discussion and comparison in the final version. Thank you again for your valuable suggestion.
>
> ## **II. Concern about More Character Generalization**
> ---
> We are grateful for your insightful questions regarding our method's scalability. We appreciate the opportunity to clarify this crucial point.
>
> About our training data, the original MultiDance dataset consists of two-person videos. This makes your question about generalization to more characters a critical test for our method. Our goal was to demonstrate that our model learns the *general principle* of identity correspondence, rather than simply remembering two-person patterns.
>
> For visual illustration, we kindly invite you to refer to **Appendix Figure 4** in our supplementary material. This figure showcases a challenging three-person scene, which serves as a 'zero-shot' test since the model was not trained on such configurations. We hope it provides qualitative evidence for the robustness of our method.
>
> However, to make this evaluation more comprehensive and to further address your concern, we constructed a new, challenging benchmark as a supplement to our original ICE benchmark. This new benchmark contains approximately 2,000 frames containing complex interactions between **3 to 5 characters**. On this new benchmark, our method demonstrates significant and consistent improvements, quantitatively confirming its scalability:
>
> | Method | PSNR*↑ | LPIPS↓ | SSIM↑ | FID↓ | FVD↓ | FID-VID↓ |
> | :--- | :--- | :--- | :--- | :--- | :--- | :--- |
> | AnimateAnyone | 14.70 | 0.370 | 0.606 | 56.66 | 401.51 | 35.356 |
> | AnimateAnyone*| 14.92 | 0.363 | 0.607 | 64.34 | 406.08 | 36.308 |
> | UniAnimate | 15.62 | 0.338 | 0.640 | 57.75 | 348.91 | 29.030 |
> | **Ours** | **16.68** | **0.315** | **0.671** | **42.84** | **261.01** | **23.571** |
>
> Furthermore, a key advantage of our framework is that this scalability comes with **no additional** computational overhead at inference time, as the Identity Matching Graph (IMG) is only constructed during training.
>
> In summary, both our original qualitative evidence and these new quantitative results on a challenging 3-5 person benchmark support our method's generalization performance. We will include these new results and benchmark to our final version.
>
> ## **III. Concern about Single-Character Generalization**
> ---
> Thank you for this insightful suggestion. Your concern about our multi-character fine-tuning might affect single-character performance is reasonable. We found that our method achieves an overall performance improvement on the single-character benchmark, and we conducted an ablation study to verify the reason of this improvement. The experiments were conducted on the **public** TikTok (Jafarian et al., 2021) benchmark.
>
> | Method | PSNR*↑ | LPIPS↓ | SSIM↑ | FID↓ | FVD↓ | FID-VID↓ |
> | :--- | :--- | :--- | :--- | :--- | :--- | :--- |
> | Moore-AnimateAnyone | **17.85** | 0.280 | 0.768 | 52.15 | 209.14 | 25.864 |
> | AnimateAnyone* | 17.19 | 0.291 | 0.764 | 62.52 | 213.68 | 25.943 |
> | w/ IEG | 17.41 | 0.289 | 0.766 | 51.52 | 195.42 | 24.515 |
> | End2End | 17.65 | 0.288 | 0.770 | 45.25 | 186.34 | 22.515 |
> | End2End-M | 17.62 | 0.285 | 0.769 | 43.17 | 182.16 | 22.881 |
> | **Ours** | 17.78 | **0.279** | **0.772** | **40.56** | **163.85** | **20.294** |
>
> Our analysis is as follows:
>
> * First, simply fine-tuning the backbone on our multi-character dataset (`AnimateAnyone*`) leads to a performance drop on the single-character task.
> * To isolate the effect of different driving pose guidance, we fine-tuned the backbone using only our Identity-Embedded Guidance (`w/ IEG`), which is slightly better than `AnimateAnyone*`.
> * Next, both `End2End` and `End2End-M` introduce reference IEG of the reference image for labeling identity (see our Appendix B.4 for details). The results show that introducing IEG for identity guidance is beneficial.
> * Finally, our full method (`Ours`) achieves the best overall results. This demonstrates the effectiveness of our **Identity Matching Graph (IMG)**.

---

> > ### Comment · Reviewer_acUv · 2025-08-06
> >
> > Thanks a lot for the rebuttal. The rebuttal well addressed some of my concerns. However, although the authors provide the experimental comparison as well as the fundmental differences, I would still argue the proposed approach has similar motivation against Follow-Your-Pose v2. Also, the authors provide more evaluations based on different number of people setting. However, the metrics used in the experiments normally reflect part of the performance of the proposed algorithms but may not be fully consistent with human preference. Thus, I would maintain my original score.

---

> ### Author Response · Authors · 2025-08-06
>
> Dear Reviewer,
>
> Thank you very much for your continued engagement and discussion. We sincerely respect your final decision and appreciate your valuable feedback. However, we would like to respectfully clarify two points:
>
> **Regarding similar motivations**: While both works target multi-person scenarios, they address different settings with distinct approaches. Follow-Your-Pose v2 explicitly mentioned that it does **not** handle character swapping scenarios, whereas our work specifically tackles this challenging problem. Our method explicitly models identity correspondence and provides direct supervision for intermediate generation processes. In contrast, Follow-Your-Pose v2 introduces a Depth Order map as an additional conditional input, which is an implicit end-to-end training approach. This fundamental difference in methodology reflects our different motivations.
>
> **Regarding human preference**: In **Appendix F**, we conducted a user study that includes feedback from real human evaluators across three dimensions, particularly the IC Accuracy dimension. We would be happy to provide additional details about our human evaluation if you would find them helpful.
>
> Thank you again for your insightful feedback and for taking the time to engage with our work.
>
> Best regards,
>
> The Authors

---

> > ### Comment · Reviewer_acUv · 2025-08-07
> >
> > Thanks a lot for the comments. I would appreciate the clarifications for the motivations as well as the human preference study. I would take these comments into the final rating.

---

> > > ### Author Response · Authors · 2025-08-09
> > >
> > > Dear Reviewer,
> > >
> > > Thank you again for your valuable suggestions and your open attitude towards our clarifications on the two further concerns regarding motivation and human preference. We greatly appreciate your openness throughout this discussion.
> > >
> > > We would like to know if there are any other concerns that we could address for you? If we have addressed most of your concerns, we would be grateful if you could share your final decision based on these clarifications.
> > >
> > > Thank you once more for your time and active engagement in this discussion.
> > >
> > > Best regards,
> > >
> > > The Authors

---

> ### Author Response · Authors · 2025-08-07
>
> Dear Reviewer,
>
> We sincerely appreciate your positive engagement and constructive feedback throughout this review process, as well as your openness to our clarifications.
>
> Should you have any further questions or concerns about our work, we would be more than happy to provide additional clarifications or details.
>
> Thank you again for your time and valuable insights.
>
> Best regards,
>
> The Authors

---

### Author Response · Authors · 2025-08-04

Dear Reviewers,

We would like to express our gratitude for your insightful comments and constructive suggestions. Your feedback is really helpful for us. We conducted additional experiments and provided responses to try to address your concerns.

We fully understand that this is a busy period for you, as you may be reviewing rebuttals for multiple papers. We would sincerely appreciate it if you could take some time to review our responses and let us know whether we have addressed all your concerns.

Thank you once again for your time.

Best regards,

The Authors

---

### Decision · Program_Chairs · 2025-09-17

**Decision:**

Accept (poster)

**Comment:**

After the rebuttal, all reviewers are consitently positive. Thus, it is a clear acceptance. Since many experiments was conducted during rebuttal period, authors need to include them in the final version.